# APPROACHING AN UNKNOWN COMMUNICATION SYSTEM BY LATENT SPACE EXPLORATION AND CAUSAL INFERENCE

## ABSTRACT

This paper proposes a methodology for discovering meaningful properties in data by exploring the latent space of unsupervised deep generative models. We combine manipulation of individual latent variables to extreme values with methods inspired by causal inference into an approach we call *causal disentanglement with extreme values* (CDEV) and show that this method yields insights for model interpretability. With this, we can test for what properties of unknown data the model encodes as meaningful, using it to glean insight into the communication system of sperm whales (*Physeter macrocephalus*), one of the most intriguing and understudied animal communication systems. The network architecture used has been shown to learn meaningful representations of speech; here, it is used as a learning mechanism to decipher the properties of another vocal communication system in which case we have no ground truth. The proposed methodology suggests that sperm whales encode information using the number of clicks in a sequence, the regularity of their timing, and audio properties such as the spectral mean and the acoustic regularity of the sequences. Some of these findings are consistent with existing hypotheses, while others are proposed for the first time. We also argue that our models uncover rules that govern the structure of units in the communication system and apply them while generating innovative data not shown during training. This paper suggests that an interpretation of the outputs of deep neural networks with causal inference methodology can be a viable strategy for approaching data about which little is known and presents another case of how deep learning can limit the hypothesis space. Finally, the proposed approach can be extended to other architectures and datasets.

## 1 INTRODUCTION

How do we approach a communication system for which we not only do not understand what is meaningful but are also unsure about how to go about testing for meaning? One such case is the communication system of sperm whales (*Physeter macrocephalus*), whose vocalizations very likely carry meaning because they are produced in duet-like exchanges or group choruses (33), but never while alone (38), and appear to be socially learned (29). While evidence supports the use of codas as identity signals (16), there is little that is currently known about what individual utterances mean, or even concrete evidence about what kind of properties of the communication system could serve to carry meaning.

In this paper, we propose an approach that aims to provide insight into the latter by using an expressive generative model as the learning mechanism. The network is trained with two objectives: (i) imitation of data and (ii) encoding of information (Fig. 2a). We then combine latent space exploration with methods borrowed from causal inference into a methodology we call *causal disentanglement with extreme values* (CDEV, Fig. 2b). This approach helps us test for what observable properties the network has learned encode uniquely relevant information for the synthetic vocalizations. If sperm whales encode information into their vocalizations and our model can learn to imitate those well, the encoding in our models can likely reveal what might be meaningful in the sperm whale communication system. We argue that our technique reveals both properties that were posited as meaningful by

human researchers as well as novel properties that have, so far, not yet been hypothesized as such. The former can thus serve to bolster the credibility of the latter.

The advantage of the proposed approach is that the discovery of such properties uses as few assumptions as possible, treating the network as a black-box learning unit. The fiwGAN architecture (4) used has been repeatedly shown to learn a variety of meaningful properties about human language from audio recordings of human speech. Networks trained on raw speech data are shown to learn to associate lexical items with code values and sublexical structure with individual bits (4; 6), uncovering known linguistic rules (3; 5; 2). In other words, the method is verified on the human communication system— language —where the ground truth is available. We observe the same ability to infer the hidden structure of the data in our results (Sec. 4). Moreover, learning is performed on raw audio without information-losing (or biasing) transformations.

To our knowledge, this is the first attempt to model sperm whale communication with deep learning on raw acoustic data. In this context, we use the network (in conjunction with CDEV) as an extraordinarily flexible and information-preserving tool for decomposing the data into meaningful, observable properties. Furthermore, the proposed approach, where individual units in the inputs of deep neural networks are manipulated to extreme values, and their effects on observable properties of the outputs are estimated via causal inference methods, could also be applied to other architectures and data with few, if any, modifications.

## 1.1 SPERM WHALE COMMUNICATION

Sperm whales communicate in short (less than two seconds), socially learned series of *clicks* called *codas* that marine biologists have grouped into several coda *types* based on the variation in the number, rhythm, and tempo of the clicks ((36; 37) (Fig. 1). In the dataset used, there are about 22 different coda types (15). The actual distribution of coda types is highly asymmetrical - the two most common codas comprise 65% of all coda vocalizations recorded (16).

Due to this rarity of some types, we further limit the training set used in this paper to the five most common coda types (1+1+3, 5R1, 4R2, 5R2, and 5R3). Since we are dealing with a communication system, a significant portion of the data could not be used due to too strong of a presence of whale dialogue (33). The residual effects of this are dealt with by our detection algorithms (Sec. 2). Restricting the number of coda types also allows us to test whether the network can predict unobserved coda types. All in all, 2209 distinct training samples were used. GANs, contrary to other architectures, do not necessarily require extensive datasets and have been shown to learn informative properties of language with similar or substantially smaller datasets (5; 4).

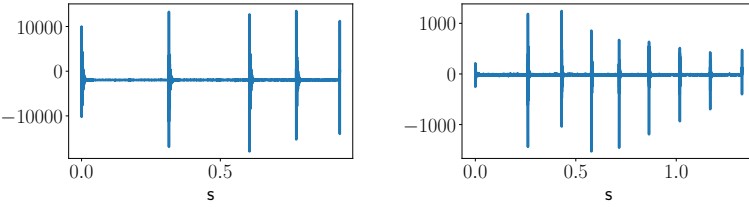

Figure 1: Examples of real codas. **Left**: type *1+1+3* with 5 clicks, **right**: type *9R* with 9 clicks. The 9R type was never part of the training data, yet our network learns to generate codas that resemble this type (Sec. 4).

## 1.2 RELATED WORK

Combining latent space exploration with the use of causal inference estimators on generated audio trained on data for which we have no ground truth is a novel approach, borrowing both from machine learning interpretability methods, as well as disentangled representation learning (DRL) (7).

In terms of the former, partial dependence plots (13) sample a section of the data randomly while examining the predicted outcome of a supervised model, which bears some similarities to our application of the ATE estimator (Sec. 3). Local surrogate models, such as LIME and SHAP (30; 25),

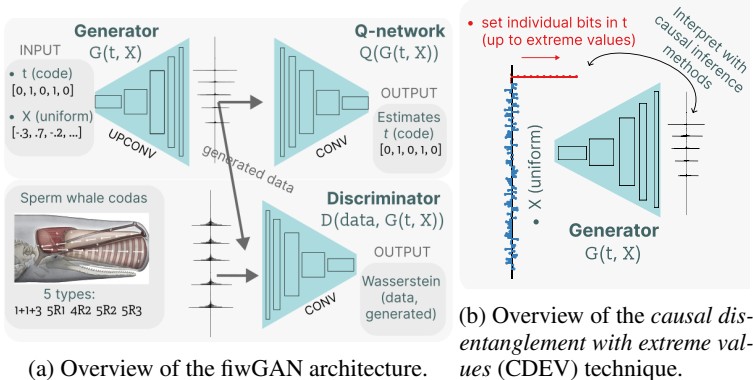

(a) Overview of the fiwGAN architecture.

(b) Overview of the *causal disentanglement with extreme values* (CDEV) technique.

Figure 2: Model and approach overview.

instead focus on explaining the local effect of perturbations via a surrogate model. While both methods bear some similarities to our application of the ICE estimator (Sec. 3), they are based on individual predictions.

In terms of the taxonomy of disentangled representation learning presented in (35), our method is related to dimension-wise unsupervised DRL due to the use of an InfoGAN (10) variant as the learning mechanism. We eschew any assumptions about the structure of the generating factors (22), as well as modifying the loss function of the architecture itself; both of these are easier to justify in applications where the ground truth exists, e.g., image generation (17; 24). Additionally, the application to audio data is a relatively unexplored case; these two factors prompt us to base our approach on an architecture (4) that has been shown to learn meaningful representations on the closest dataset where the ground truth is available - human speech, and "perform the disentanglement *post-hoc*" (i.e., from generated data) with the use of causal inference methods. Fully causal DRL methods (34) often use a structural causal model as a prior (39), which is, again, hard to justify for an unknown communication system. As an example, (9; 8) use causal inference to quantify implicit causal relationships *between* latent space variables, while our use is based on effect estimation in service of uncovering encoded properties. As an example of the latter, (11) use causal methods for interpreting CNNs in a supervised classification setting when the input is occluded.

## 2   THE ARCHITECTURE USED AND THE CDEV METHODOLOGY

The network architecture used in this work is fiwGAN (4), an InfoGAN (10) adaptation of the WaveGAN (12) model. In short, the input is partitioned into an *incompressible noise* $X$ and an additional *featural encoding* $t$, which is sampled as Bernoulli variables during training (4). The goal of the additional Q-network during training is to correctly determine the value of the encoding $t$ while only having access to the generated data; a loss based on the mutual information between the two is backpropagated to the generator otherwise. This ensures consistency of output across similar values of the encoding vector (Fig. 2a) and encourages disentanglement in a game that mimics communicative intent - learning by imitation in a fully unsupervised manner.

Beguš (2) proposes a technique to uncover individual latent variables that have linguistic meaning by setting them to extreme values outside of those seen in training and interpolating from there. In this work, we conversely treat the generator as an experiment in the vein of causal inference and test for *observable* properties of the data that have (or can be) hypothesized as meaningful. When generating output samples, the incompressible noise - $X$ - entries are sampled randomly, while the featural encoding $t$ is set manually to a desired value. Since the consistency of output with regard to the encoding is only enforced in a loose way, this relationship often only becomes readily apparent when setting the numerical values outside the bounds seen in training, where the primary associated effect begins to dominate (2; 3; 5). We then apply statistical estimators on the candidate property samples derived from the raw generated outputs to determine whether there exists a statistically consistent relationship between the encoding and the outcomes. This procedure gives rise to a methodology we call *causal disentanglement with extreme values* (CDEV) (Fig. 2b).

The space available for the featural encodings is limited; hence, finding the real-world attributes that map almost one-to-one with the encodings suggests that the generator considers them very important to generating convincing outputs, in which it is being checked by the discriminator with access to real data. We limit our featural encoding space to five bits ($= 2^5 = 32$ classes) for the five coda types present in the data to allow the model to capture compositionality. However, we demonstrate in Appendix A.1.1 that the method is robust to the number of bits chosen, as well as model training specifics. Similarly, on language data, the architecture uncovers meaningful properties even when there is a mismatch between the number of true meaningful classes and the size of the binary code (4). Therefore, any intuition about the desired size of the encoding acts only as a rough prior. Additionally, the uniqueness of the encoding matches is verified by an unrelated method, presented in Appendix A.7.

For the first two observables considered - the number and regularity of clicks (Sections 3 and 4) — we use an algorithmic *click detector*. As we move towards more extreme encoding values, the output becomes progressively noisier, as shown for a relatively pathological example in Figure 3. Therefore, the detector uses minimal thresholds, signal filtering, and a sub-algorithm that, if multiple valid "solutions" given the minimal threshold constraints are still found, selects the more evenly spaced-out solution, which corresponds to our intuition. In concert with this, we chose the upper limit of the range tested ($t = 12.5$) so that the quantities measured by algorithms remained consistent despite increasingly noisy output (details in Appendix A.4). The latter might, in part, be due to the network training not having converged completely; since the goal is to discover what observable properties are encoded and not high-fidelity whale communication generation, this is not a significant impediment. Likewise, the actual numeric magnitudes of the estimated effects are less important than discovering the existence of such consistent relationships. The fact that the disentanglement might not be possible for some observables where the detection is too noise-sensitive, together with the need for sufficient sample sizes, which necessitates algorithmic measurement being possible and (potentially) incurs a considerable computational cost, thus act as the main limitations on the proposed method besides the need to specify candidate properties to test for.

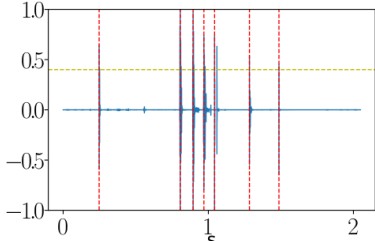

Figure 3: Example of the application of the click detector on noisy generated data. The minor peaks are artifacts picked up in training due to the data not being free of whale dialogue, the red vertical lines are the detected clicks, and the yellow horizontal line is the inferred volume level for the vocalization to be coming from the primary whale.

## 2.1 THE EXPERIMENT FROM THE CONTINUOUS-TREATMENT CAUSAL INFERENCE POINT OF VIEW

The methodology used in this work is inspired by *continuous-treatment* causal inference from the Neyman-Rubin framework point of view (32). In short, it deals with an experimental setup where the usual treatment assignment variable $Z$ becomes a continuous variable $T$, often called a *dose* due to it being common in the pharmacological context. Given the assumptions common to the discrete case, the analysis proceeds in much the same fashion, with the treatment dose being discretized at several levels $t$. In the case of observational studies, the *propensity score* $e(x)$ thus becomes a function of two variables $r(t, x)$ (19):

$$e(x) \rightarrow r(t, x) = \mathbb{P}(T = t | X = x);$$

in any setup, the most common quantity of interest is again the difference in the conditional expectations of the outcome $Y$:

$$\mathbb{E}_{Y(t)}[Y(t)|r(t, X) = r] - \mathbb{E}_{Y(s)}[Y(s)|r(t, X) = r],$$

relative to some *baseline dose* $s$. The choice of the latter is natural ($s = 0$) in pharmacology, for example, where no drug being administered carries a special meaning. In our context, the choice of such a baseline is not clear, as will be addressed in this work. The ultimate goal is to obtain the full *outcome curve* for all $t$ with regard to the $s$ chosen.

In terms of causal inference terminology, we have the following variables: the *incompressible noise* part of the output is treated as the 95-dimensional covariate vector $X$, while the *featural encoding* is considered to be the treatment $t$ and is a five-dimensional vector. The observables considered, such as the number of clicks for a given $X$ and $t$, are thus the outcome variable $Y_i$. Each output was generated by sampling $X \overset{\text{iid}}{\sim}_{1\ldots95} \sim Unif[-1, 1]$. Such vectors can be treated as unique within the generated sample used for the estimation of a particular quantity and hence can be thought of as denoting a separate *unit* indexed by $i$ (cf. Appendix A.8). These were fed to the generator at each level of the treatment $t \in [-1, 12.5]$ set at each of the five featural bits, meaning the covariates $X$ were kept the same across treatment levels. The generated audio was then run through the appropriate detection algorithm, such as the click detector, giving us the outcome $Y_i(t)$. The total number of units was chosen as $N = 2500$, for which we determined the estimates to have completely stabilized (cf. Appendix A.1.2). This means we perform a completely randomized experiment with the added bonus that the outcome is observed at each treatment dose for each unit, meaning that there is no *fundamental problem of causal inference* (18) at play here, simplifying the estimation.

## 3    AN INTRODUCTION TO THE METHODOLOGY: THE NUMBER OF CLICKS

The first observable we're interested in is the *number of clicks* in the generated audio samples. As this quantity is the easiest to visualize, this section will also introduce the estimators used. The first estimator used is the *average treatment effect* (ATE), applied separately at each treatment dose value $t$. Formally, we're interested in the effect relative to some *baseline dose* $t'$:

$$\mathbb{E}[Y(t)] - \mathbb{E}[Y(t')] = \mathbb{E}_X[\mathbb{E}_Y[Y|X, T = t]] - \mathbb{E}_X[\mathbb{E}_Y[Y|X, T = t']] \tag{1}$$

Since the units $i$ are defined by their randomly drawn $X$, and we observe every $Y_i(t)$, this simply corresponds to the difference in sample averages:

$$\hat{\tau}(t) := \frac{1}{N} \sum_{i=1}^{N} Y_i(t) - \frac{1}{N} \sum_{i=1}^{N} Y_i(t'), \tag{2}$$

The treatments correspond to setting the values of single bits in the encoding while keeping the others at zero. This leaves us to consider what would be the most natural value for the baseline; the limits of the training range: -1, where the output coalesces out of pure noise (cf. Appendix A.4) and +1, where the process of disentanglement begins, serve as logical choices; the results are shown in Figure 4.

We observe a high degree of entanglement (4) of the learned encodings within the range seen in training ($[-1, 1]$). However, when setting the treatment value above that range, all the bits but *bit 1* stabilize at roughly a constant effect, which lends credence to interpreting this as a *process of disentanglement*: with higher values, the primary encoded effect in each bit starts to dominate, leading the other bits to lose their secondary effects on the number of clicks. Thus, we propose that *bit 1* primarily encodes the number of clicks. In addition to the effect of bit 1, we also note a persistence of an effect in bit 3, which stabilizes at a different stationary value. Examining the standard deviation in the number of clicks, as shown on the bottom of Figure 4, we again observe the same phenomenon of *disentanglement*, leading us to conclude that *bit 1* encodes the *range of clicks* output by the generator.

The question of selecting the appropriate baseline dose is elegantly avoided by using the *incremental causal effect* (ICE) proposed by (31) — the effect due to an infinitesimal shift in dosage:

$$\tau_{ICE} := \mathbb{E}[Y(T + \delta)] - \mathbb{E}[Y(T)]. \tag{3}$$

Since we are dealing with a completely randomized experiment (Sec. 2.1, Appendix A.8), the following holds:

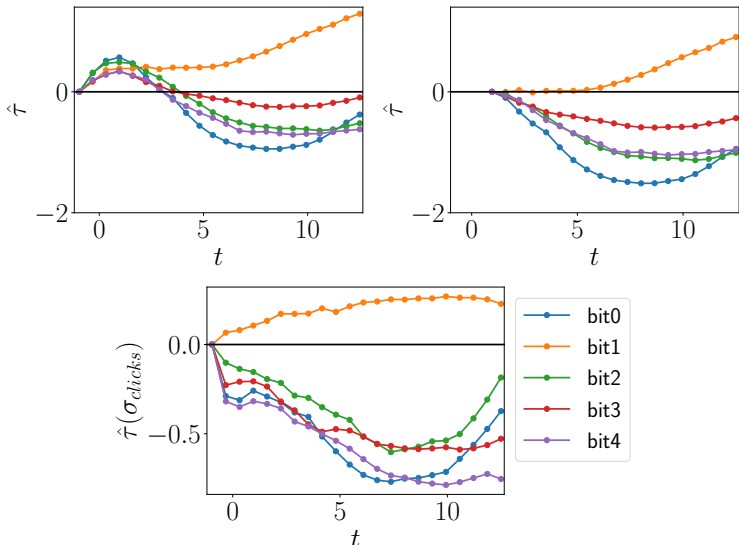

Figure 4: **Top:** The ATE for the number of clicks with the baseline chosen as **left**: at $t' = -1$, **right**: at $t' = 1$. **Bottom:** The ATE for the standard deviation in the number of clicks for the baseline at $t' = -1$.

$$\partial_t \mathbb{E}[Y|T = t, X = x] = \mathbb{E}[Y'(t)|T = t, X = x], \qquad (4)$$

meaning that the left-hand side - the true incremental causal effect - is identifiable by the expectation of the derivative - the right-hand side. Its estimator — $\hat{\tau}_{ICE}$ — is the usual sample mean of the numeric finite differences in the outcomes for single units. Given the expectation of the derivative curve (Appendix A.6), we can also define the *expected effect of an infinitesimal increase in the treatment:*

$$\hat{\theta}_{fs} := \frac{1}{N_t} \sum_{i_t=1}^{N_t} \mathbb{E}_{Y'(t_{i_t})}[Y'(t_{i_t})|T = t_{i_t}, X = x_i] = \frac{1}{N_t} \sum_{i_t=1}^{N_t} \frac{1}{N} \sum_{i=1}^{N} \frac{\Delta Y_i(t_{i_t}, x_i)}{\Delta t_{i_t}}, \qquad (5)$$

where $N_t$ is the number of examined treatment levels with the corresponding $t_{i_t}$, and $N$ is the number of units with the corresponding $x_i$, each receiving each treatment level. In short, this is the *expected overall effect* if all the current treatments were infinitesimally shifted (in the positive direction) for the given sample. As a single metric, it can be interpreted as an analog to the *average treatment effect* for binary treatments without the need for a baseline dose as a reference. Its values are presented in Table 1, corroborating the results obtained with the average treatment effect estimator.

Table 1: The expected effect of an infinitesimal increase in the treatment on the number of clicks.

| bit | 0 | 1 | 2 | 3 | 4 |
|---|---|---|---|---|---|
| $\hat{\theta}_{fs}$ | -0.028 | 0.096 | -0.039 | -0.007 | -0.046 |
| $\hat{\theta}_{fs}\|t \geq 1$ | -0.082 | 0.079 | -0.087 | -0.038 | -0.083 |

## 4 CLICK SPACING AND REGULARITY

We can also apply this methodology to test for additional properties of the communication system that the network encodes as meaningful. The observables we are interested in in this section are the *click spacing*, as measured by the mean *inter-click interval* (ICI) of a coda, and the *coda regularity*, which we measure by the standard deviation of the inter-click intervals within a coda. Since these

quantities are not completely independent of the overall coda length, we stratify the results by the number of clicks observed.

The top row of Figure 5 shows the ATE with regard to the baseline at $-1$ for the mean ICI for bits 1 and 2. We observe a consistent, monotonic effect in the value of bit 1, while bit 2 (and others not shown in the figure) do not display any such consistency across varying numbers of clicks.

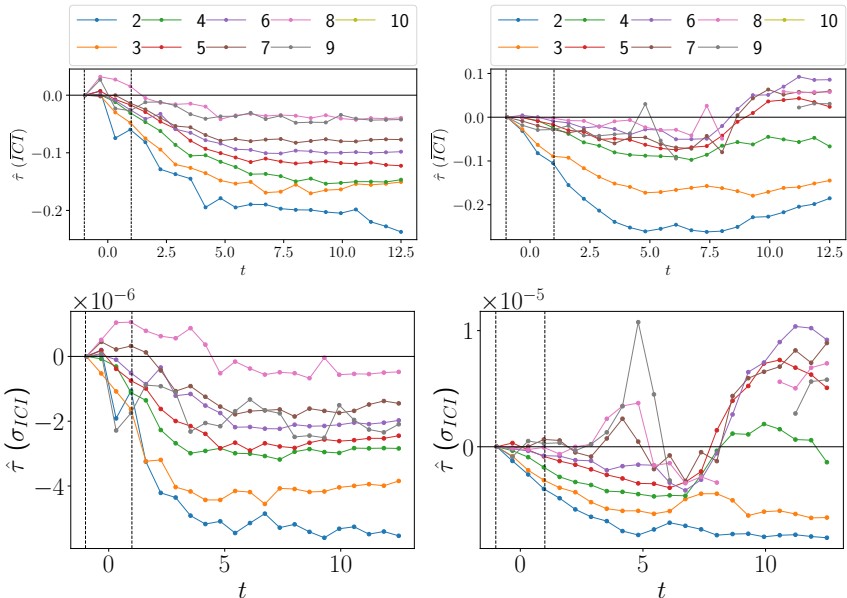

Figure 5: **Left:** Bit 1, **right:** bit 2. **Top: Mean ICI ATE**, stratified by the number of clicks. **Bottom: Coda regularity ATE**, stratified by the number of clicks. The training range is denoted by dashed lines, with the baseline set at $t' = -1$ for all.

We present the ATE for coda regularity in the bottom row of Figure 5 for the baseline at $t = -1$, again for bits 1 and 2. We observe that bit 1 additionally encodes an increasing *coda regularity* (i.e., decreasing variance in the spacings between clicks) across all coda types (proxied here by the number of clicks). Since it also encodes the number of clicks, this implies that the generator has learned to connect these properties: the codas with a higher number of clicks are more regular, with the clicks being more closely spaced together. This is especially poignant since the same property holds for actual whale codas. Gero et al. (15) suggest that as coda length in clicks increases, mean ICI decreases to fit clicks within this duration limit, which appears to be the result of avoiding overlapping with the next coda within an exchange between whales. Furthermore, codas with more clicks are often more regular in their ICIs, regardless of their duration. The generator has thus inferred this connection *without the codas with a high (>5) number of clicks even being present in the dataset* due to data quality limitations (cf. Sec. 1.1) and encoded it in the limited space (five bits) it has reserved for encodings. In this, we again observe the remarkable propensity of generative adversarial networks to discover hidden structures of the data and innovate in semantically meaningful ways.

As a summary of the overall incremental effects, we can apply and sum the *sign* function on the expected effect of an infinitesimal increase in the treatment across codas with varying numbers of clicks, which we call the *sign score* and present in Tables 2 and 3. This metric confirms that bit 1 is consistent in its encoding of both click spacing and regularity, regardless of the overall number of clicks in the generated codas. While bit 3 again remains slightly entangled for the latter quantity, its actual values of the expected effects were smaller than those of bit 1 (cf. Appendix A.6.1).

## 5 ACOUSTIC PROPERTIES

We now apply the methodology to acoustic quantities, as captured by the spectra at the click or coda level. So far, little is known (or has been hypothesized) about the informational content of the

| bit | 0 | 1 | 2 | 3 | 4 |
|---|---|---|---|---|---|
| *sign score* | 4 | -10 | -4 | -4 | -4 |

Table 2: Overall *sign score* of an infinitesimal increase in the treatment on the mean inter-click distance.

| bit | 0 | 1 | 2 | 3 | 4 |
|---|---|---|---|---|---|
| *sign score* | 6 | -10 | -2 | -8 | -2 |

Table 3: Overall *sign score* of an infinitesimal increase in the treatment on the inter-click interval standard deviation.

acoustic properties of whale communication. The first quantity we consider is the *mean spectral frequency*, where we first isolate the clicks with the click detector, estimate the mean frequency of each periodogram, and compute the average across all the per-click means in a particular generated coda. The corresponding ATE is displayed on the top left in Figure 6.

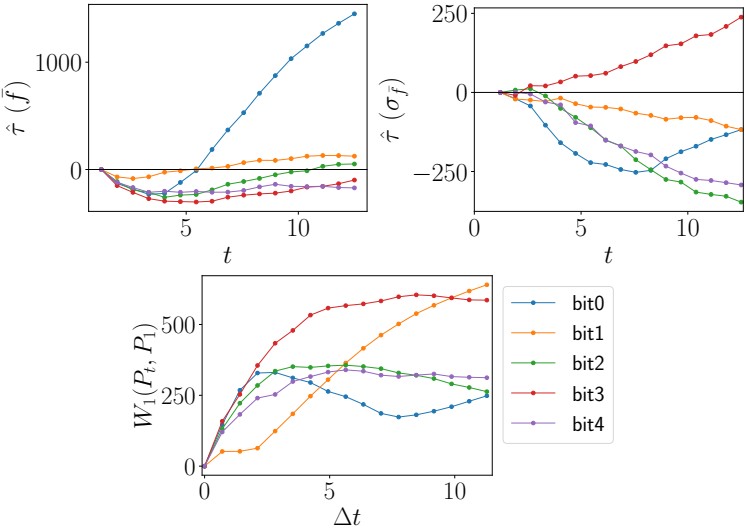

Figure 6: **Top**, **left**: the ATE for the click-level *spectral mean* with the baseline at $t = +1$, **right**: the ATE for the click-level *spectral mean standard deviation* with the same baseline. **Bottom:** The Wasserstein mean coda-level spectral distances to the same baseline.

We again observe the familiar pattern corresponding to *CDEV — bit 0* has a positive effect on the observable quantity while all the rest converge to a smaller, stationary value. Following the outlined process, we thus hypothesize that this bit — *bit 0 — encodes the spectral mean*. The potential for the spectral means of codas to be meaningful has not been hypothesized in previous research, but the hypothesis is not completely ungrounded. In a recent paper, (27) suggest that odontocetes can vocalize in different registers and that their articulators are sufficiently flexible for such differences to be possible.

We are also interested in *acoustic regularity*, which we measure by the standard deviation of the click spectral means *within* each generated coda. The results are presented on the top right in Figure 6. The results suggest *an additional meaningful encoding: bit 3* appears to encode the within-coda *acoustic regularity* of the output. In this case, it can be suspected that bits 1, 2, and 4 have not yet fully reached stationary values of their disentanglement.

Table 4 presents the results for the expected effect of an infinitesimal increase on the click mean frequency, both for the overall range of treatments and the range above the training range. The results corroborate the results obtained via the ATE estimator with the effect of bit 0 dominating the others. Similarly, the results for coda acoustic regularity in Table 5 show bit 3 to be the outlier, with bits 0 an 1 (for which we have uncovered associated observable effects, hence likely having an "unintended" effect on the spectral mean standard deviation) also being relative outliers to the baseline of *disentanglement* evident in bits 2 and 4.

| bit | $\hat{\theta}_{fs}(\bar{f})$ | $\hat{\theta}_{fs}(\bar{f})|t \geq 1$ |
|---|---|---|
| 0 | 62.15 | 128.58 |
| 1 | -30.07 | 10.99 |
| 2 | -44.55 | 4.56 |
| 3 | -45.48 | -8.71 |
| 4 | -53.95 | -15.18 |

Table 4: The expected effect of an infinitesimal increase in the treatment on the average click mean frequency.

| bit | $\hat{\theta}_{fs}(\bar{\sigma}_f)$ | $\hat{\theta}_{fs}(\bar{\sigma}_f)|t \geq 1$ |
|---|---|---|
| 0 | -21.17 | -10.42 |
| 1 | -21.99 | -10.42 |
| 2 | -35.85 | -30.74 |
| 3 | 7.37 | 21.09 |
| 4 | -34.01 | -21.83 |

Table 5: The expected effect of an infinitesimal increase in the treatment on the click mean frequency standard deviation within a coda.

Since the spectra are distributions themselves, we can measure the effect of specific bits on the *overall acoustic output*, as captured by the average coda-level spectrum (across $N$ units with random $X$ for each fixed $t$). The Wasserstein distances from the baseline at $+1$ of the average coda-level spectra are presented on the bottom in Figure 6. The result shows a relative spectral stabilization close to the limit of the training range for bits 2 and 4. Notably, bits 0, 1, and 3, for which we have uncovered observable effects - spectral mean, click number (range) and regularity, and spectral regularity, respectively, continue to evolve their average spectra with further treatment.

## 6 CONCLUSION

In this paper, we have presented a model-agnostic approach to test for candidate observable properties a deep generative model encodes as meaningful as a way of gleaning information from data that is alien to us in the true sense of the word: the communication of sperm whales. In this, we leverage the power of information-theoretic GANs to encode semantically meaningful properties in a completely unsupervised fashion. Since the model is constrained in the number of such encodings it can learn, we can argue that it must consider these critical to its ability to generate data.

To uncover these properties, we consider the trained model as an experiment and propose a method we call *causal disentanglement with extreme values* to discover the encodings. We present causal inference-inspired estimation methods that enable us to consistently pair up particular bits of the encoding with a physically observable property of the communication system. The agreement between the methods gives further credence to the results.

With this setup, we confirm that the number of clicks, which is what the existing classification developed by marine biologists is primarily based upon, indeed seems to be a fundamental property of the communication system. This can be seen as a good grounding point with regard to the credibility of the approach. While generating innovative examples of codas not seen in the data, we discover that the network correctly associates synthetic codas with a high number of clicks with their increasing regularity by encoding both properties simultaneously. Thus, it correctly infers a property of unseen real-life codas, illustrating its ability to learn the hidden structure of the data. Using the proposed technique, we also uncover that two acoustic properties might be meaningful in the system: (i) the coda spectral mean and (ii) the regularity in the spectra across the clicks within a coda (coda spectral regularity). These two properties have not been considered significant by previous research. This could serve as an indicator that the communication system is not merely a Morse-like code where only the number of clicks and the intervals between them are meaningful, but that whales actively control the acoustic properties of their vocalizations and encode meaning in those properties.

Finally, the methodology presented could be applied to any problem for which one would like to leverage the immense expressiveness of deep generative models to consistently test for what properties of the data are semantically meaningful.

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

## A APPENDIX

### A.1 THE REPOSITORY, REPRODUCIBILITY, AND ROBUSTNESS

The accompanying repository is included in the accompanying zip file. It contains the source code for the model used as the base learning mechanism, the experiment data generating code, and the analysis code needed to replicate the results presented in the paper.

Since we cannot provide the raw audio source data (cf. Appendix A.3), we provide trained models at https://zenodo.org/records/10157035. The two models included therein are the main model with an encoding space size of five bits, as well as an independently trained model with an encoding space size of six bits that is used to demonstrate the reproducibility of the results in A.1.1.

We also provide intermediate generated data needed to reproduce the final results presented in the paper. The README.md document located at the root of the accompanying zipped directory provides instructions for reproducing the results as well as further technical details, such as the compute resources used to train the model and produce the results.

#### A.1.1 ROBUSTNESS TO ENCODING SIZE AND MODEL INITIALIZATION: RESULTS OBTAINED FROM A SEPARATE MODEL WITH A SIX-BIT ENCODING

We can verify the robustness to the choice of the number of encodings desired (i.e., size of $t$), as well as to re-initialization by performing the same analysis on an independently trained model where we selected the size of the reserved encoding space $t$ to be **six** bits. The results for the number of clicks are presented in Figure 7 and again show the same pattern of disentanglement with a single bit - bit 0 in this case - picking up the encoding.

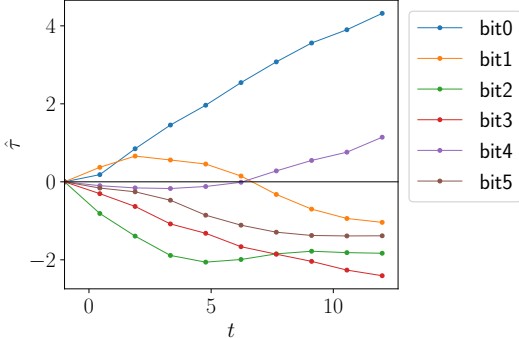

Figure 7: The ATE for the number of clicks with regard to the baseline at $-1$ obtained from a separate model that used an encoding size of **six** bits.

Please note that this bit is a different one from the one encoded by the model used for the results presented in Section 3, demonstrating that due to the nature of the unsupervised learning of the encodings, the ordering of the bits does not imply an ordering of the importance of properties. Interestingly, we again observe a degree of entanglement of the encoding with an additional bit, in this case, bit 4.

#### A.1.2 AN ESTIMATE FOR THE ACCURACY OF THE ESTIMATION: THE NUMBER OF CLICKS ATE OBTAINED FROM THE ACOUSTIC DATA

Since the raw data for the acoustic results was generated in an independent run, i.e., with the set of covariates $X$ sampled independently (with a different random seed) of those for the results for the number of clicks and their regularity presented in the paper, we can use it as a check on those results. While a complete error estimation would be computationally expensive, we present the ATEs for the number of clicks from both sets of synthetic data in Figure 8, with their almost exact match validating the significance of the results.

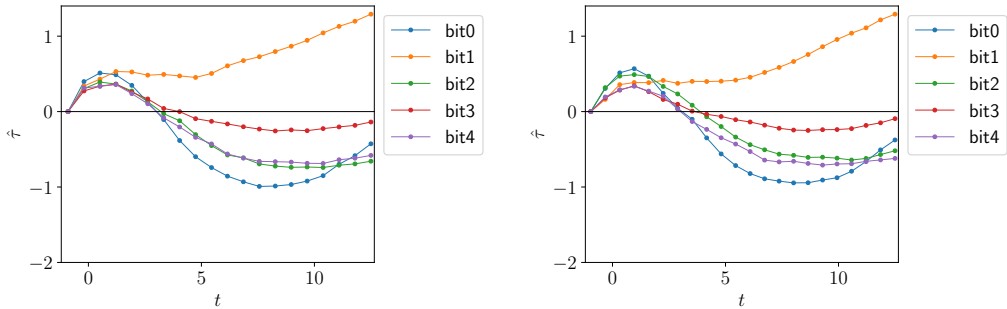

Figure 8: Comparison between the ATE for the number of clicks with regard to the baseline at $-1$ from the data generated for the **left:** acoustic results, **right:** the click results presented in the main paper.

## A.2 DETAILS REGARDING THE ARCHITECTURE

The loss opitimized by the fiwGAN architecture (4) is the following:

$$\min_{G,Q} \max_{D} V_{IWGAN}(D, G, Q) = V_{WGAN}(D, G) - \lambda L_I(G, Q), \tag{6}$$

Where $V_{WGAN}$ refers to the loss of the Wavegan architecture (12). $L_I$ is the variational lower bound of the mutual information between the encoding and the generated data $I(t; G(X, t))$, and $\lambda$ is a hyperparameter. Please see Chen et al. (10) for more details.

## A.3 THE DATA

The dataset of recordings for this study originates from The Dominica Sperm Whale Project (see Gero et al. 14) and was collected off the coast of the island of Dominica between 2014 and 2018. The vast majority of the recordings are made of one sperm whale vocal clan. In the dialect of this Eastern Caribbean Clan, there are about 22 different coda types that have been defined (15).

Codas were collected from animal-borne sound and movement tags between 2014 and 2018 (*Dtag* generation 3; Johnson and Tyack 20). *Dtags* record two-channel audio at 120 kHz with a 16-bit resolution, providing a flat (±2 dB) frequency response between 0.4 and 45 kHz. As a result, both coda and echolocation clicks produced by sperm whales were able to be recorded cleanly, allowing us to obtain the temporal patterning and spectral properties of coda clicks used in this analysis.

In order to generate the dataset used for training the neural network, the codas were extracted from the raw audio with the help of annotations provided by marine biologists and subsequently resampled to 32 kHz mono *wav* format and slightly augmented as described in the README.md document in the supplement. Finally, 2209 samples belonging to the five most common coda types (1+1+3, 5R1, 4R2, 5R2, and 5R3) were selected on the basis of the recordings being relatively free of whale dialogue.

## A.4 DETAILS REGARDING THE ALGORITHMIC MEASUREMENT OF THE QUANTITIES OF INTERESTS

While the upper bound of the range of treatments was selected so that the quantities measured by the detection algorithms remained consistent (as discussed in the main work), the lower bound was chosen as the lower bound of the training range, as all lower values (for any of the bits) induced the generator to output noise (as illustrated in Fig. 9) too often to make estimation possible.

### A.4.1 THE CLICK DETECTOR

To measure the number of clicks and the intervals between them, we use an algorithmic *click detector*. We can make use of prior knowledge to set minimum thresholds for both the amplitude and the

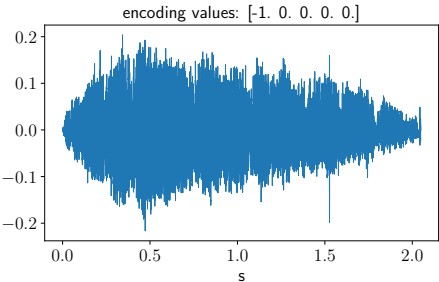

Figure 9: Example of noisy output for encoding values up to the training range limit of $-1$, in this case, $[-1, 0, 0, 0, 0]$.

temporal separation in order to choose between (potentially) multiple sets of detected "clicks". The latter threshold is based on the minimum peak separation possible by the whale physiology and was set to 40 ms. The amplitude threshold is necessary to deal with the residual presence in the dataset of interwoven codas coming from other whales; it was set to $0.4$ in terms of the relative amplitude to the peak click and obtained from an analysis of the amplitudes of vocalizations coming from the primary whale and secondary whales in the data.

Even so, the output can become progressively noisier as we move towards more extreme encoding values (in the positive direction). Therefore, the detector also uses a band-pass Butterworth filter permitting frequencies between 2 and 16 kHz (based on known spectral ranges of whale clicks), as well as a sub-algorithm that strives to "maximize entropy" by preferring well-spaced peaks over counting a single jagged peak multiple times: i.e., if multiple valid "solutions" given the minimal threshold constraints are still found, it will prefer the more evenly spaced-out solution, which corresponds to our intuition (cf. Fig. 3).

### A.4.2 ACOUSTIC PROPERTIES

The acoustic properties were derived from the raw generated audio by again applying the filter and then calculating the periodogram (computed with an added Hamming window) in the case of coda-level properties (not presented in the main paper, presented in Appendix A.5 as a supplementary result), and the spectrogram in the case of click-level properties. The latter was subsequently cut up into clicks using the click detector described above, and the quantities of interest, such as the per-click spectral mean and the per-click spectral standard deviation, obtained as the weighted statistics, with the spectrogram values serving as weights - e.g., the weighted mean of the spectrogram slice frequencies, weighted by the values of the same slice. The results presented as the click-level spectral mean and regularity were then obtained as the average of the corresponding quantities across the clicks within the coda considered.

### A.5 ADDITIONAL ATE ESTIMATOR RESULTS - ACOUSTIC PROPERTIES

In addition to the click-level quantities presented in the paper, we can also examine the same quantities at the *coda level*. The results for the mean spectral frequency ATE are presented in Figure 10 and are consistent with each other, with the coda-level estimation naturally involving more noise.

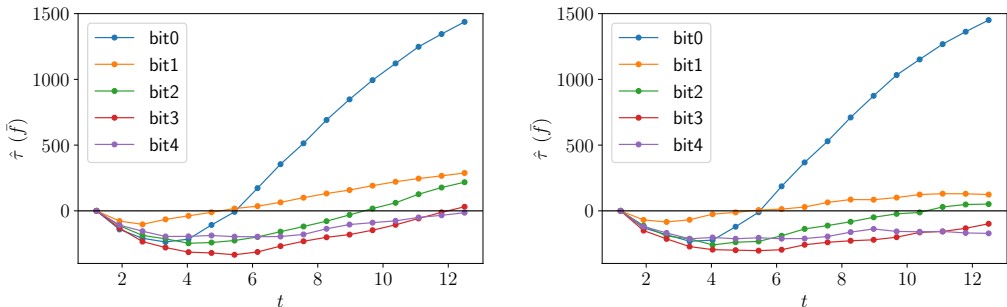

Figure 10: The (average in the case of click-level) spectral mean ATE for the baseline at $+1$ at the **left:** *coda-level*, **right:** *click-level*.

### A.6 ADDITIONAL ICE ESTIMATOR RESULTS AND DISCUSSION

Formally, Rothenhäusler and Yu (31) show that, in order for the true incremental causal effect (ICE) to be identifiable by the expectation of the derivative, we need *local ignorability* and *local overlap*. Based on the discussion in Appendix A.8, both assumptions hold trivially in our case: (i) local ignorability holds because strong ignorability holds, and (ii) local overlap holds since the probability of treatment being assigned is a constant: -1.

A potential formal drawback of using this metric in this setup is the discontinuity of the outcome for the number of clicks, which is a discrete variable. However, since the estimator used — $\hat{\tau}_{ICE}$ — is the sample mean of the finite differences, which are, naturally, always defined, we do not consider this to be a major impediment for this use case. Furthermore, this does not apply to the other observables evaluated in this work.

As an addendum to the aggregate *expected effect of an infinitesimal increase* results presented in the main text, the estimator $\hat{\tau}_{ICE}$ (i.e., the estimator for the quantity in Eq. 3) for the number of clicks is presented for two bits in Figure 11. We again observe the phenomenon of disentanglement outside the training range, where the effect of bit 1 remains consistently positive. In contrast, that of bit 4, for example, returns to an oscillation around zero.

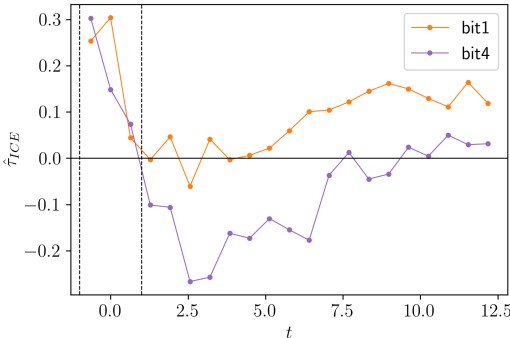

Figure 11: The incremental causal effect $\hat{\tau}_{ICE}$ estimate on the number of clicks for bits 1 and 4; the training range is delineated by dashed lines.

#### A.6.1 CLICK SPACING AND REGULARITY

Table 6 shows the expected effect of an infinitesimal increase in bit values on the mean ICI, stratified by the overall number of clicks in the coda over the range $t \in [-1, 12.5]$. These results are summarized in the *sign score* presented in Table 2 in the main text.

Similarly, we show the results for coda regularity in Table 7, summarized in Table 3 in the main text.

| # clicks bit | 2 | 3 | 4 | 5 | 6 | 7 | 8 | 9 | 10 | 11 |
|---|---|---|---|---|---|---|---|---|---|---|
| 0 | -0.013 | -0.002 | 0.007 | 0.007 | 0.005 | 0.004 | 0.004 | 0.009 | 0.008 | N/A |
| 1 | -0.018 | -0.011 | -0.011 | -0.009 | -0.007 | -0.006 | -0.003 | -0.003 | -0.003 | -0.007 |
| 2 | -0.014 | -0.011 | -0.005 | 0.002 | 0.006 | 0.004 | -0.005 | -0.010 | -0.027 | -0.008 |
| 3 | -0.017 | -0.012 | -0.006 | -0.004 | 0.000 | 0.001 | 0.003 | -0.002 | -0.008 | N/A |
| 4 | -0.022 | -0.017 | -0.011 | -0.002 | 0.004 | 0.005 | -0.001 | -0.006 | -0.001 | 0.000 |

Table 6: The expected effect of an infinitesimal increase in the treatment on the mean inter-click distance.

| # clicks bit | 2 | 3 | 4 | 5 | 6 | 7 | 8 | 9 | 10 | 11 |
|---|---|---|---|---|---|---|---|---|---|---|
| 0 | -0.004 | 0.005 | 0.023 | 0.023 | 0.019 | 0.013 | 0.016 | 0.037 | 0.019 | N/A |
| 1 | -0.013 | -0.009 | -0.007 | -0.006 | -0.005 | -0.003 | -0.001 | -0.005 | -0.007 | -0.005 |
| 2 | -0.019 | -0.014 | -0.003 | 0.012 | 0.022 | 0.021 | -0.004 | 0 | -0.072 | -0.074 |
| 3 | -0.014 | -0.009 | -0.006 | -0.003 | -0.003 | -0.001 | 0.001 | -0.013 | -0.004 | N/A |
| 4 | -0.017 | -0.012 | -0.005 | 0.012 | 0.016 | 0.016 | -0.01 | -0.006 | -0.014 | 0.015 |

Table 7: The expected effect of an infinitesimal increase in the treatment on the standard deviation of the ICIs.

Since we have shown *bit 1* to primarily encode the number of clicks, holding it at constant $t_1 = 0$ while varying the other bits results in the codas with a very large number of clicks — 11 — not being generated in the case of bits 0 and 3.

### A.6.2 Wasserstein distance ICE

We can also evaluate the expected effect of an infinitesimal increase in treatment on the average spectral Wasserstein distance (cf. Sec. 5), presented in Table 8. The clicks matched to an observable effect — bits 0, 1, and 3 – also mostly show a more pronounced expected effect on the *Wasserstein* distance between the average spectra relative to the point where the respective bit value is set to $t = 1$. The result for bit 0 is an outlier due to the fact that for the click mean frequency, the disentanglement only picks up with relatively higher values of $t$ (around $t = 5$, cf. top left of Fig. 6 in the main paper), which means that the spectrum at that point is again relatively similar to the initial point and only diverges at values above that, leading to an overall effect in the distance comparable to bits 2 and 4, which disentangle to a point with about the same average spectral distance effect.

| bit | 0 | 1 | 2 | 3 | 4 |
|---|---|---|---|---|---|
| $\hat{\theta}_{fs}(W_1(P_t, P_1))$ | 22.00 | 56.72 | 23.30 | 51.92 | 27.66 |

Table 8: The expected effect of an infinitesimal increase in the treatment on the Wasserstein distance between average spectra relative to $t = 1$.

### A.7 Regressing the outcomes directly

While our usage of the ATE and ICE estimators bears some relation to explainability methods such as partial dependence plots (13) and local methods such as LIME (30), respectively, we can also somewhat mirror the latter in another way by attempting to estimate the effects on the outcomes with a surrogate model. In practice, this means we regress the *individual* observed outcomes $Y_i(t)$ against the full input vector for each unit at each level $t$, i.e., for bit 1: $x_i = [0, t, 0, 0, 0| X_i]$. The *outcome curve* in this setting corresponds to the *mean* of the individual inferred outcomes at each treatment level.

For this task, we've chosen boosted tree regression as implemented in the `lightgbm` package (21). For our specific purpose, we intentionally use *no* sparsity-inducing regularization to prevent

overfitting: we wish to check if the hypothesized encodings are consistently picked up by a completely unrelated non-parametric method that is (i) able to approximate any function arbitrarily closely with sufficient model complexity (ii) when unregularized, is prone to overfitting. Hence, we are looking for *consistency of explanation* across varying model complexity, *despite all odds*. In other words, we are testing whether the encodings uncovered by the other two estimators could be spurious.

To explain the results and quantify the consistency of the explanations, we use feature importance values obtained from SHAP (25), which uses a game-theoretic concept called the *Shapley value* to distribute attribution to the outcome to the input features. Note that while one could, in theory, use SHAP on the generative model itself, it is much more computationally efficient when used with trees (26). As mentioned, we also wish to test our findings explicitly via an unrelated method as opposed to disassembling the generative network.

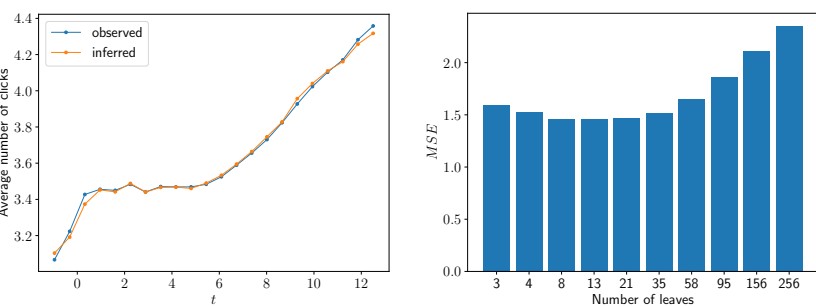

Figure 12: Regression results for the number of clicks outcome: **Left:** the outcome curve derived from the data (the mean number of clicks) and the one inferred by the best fitting model in terms of the out-of-sample MSE. **Right:** MSE by model complexity as determined by the maximum number of leaves allowed.

Most of the hyperparameters except the main one - the number of leaves - were chosen by an early stopping of the training as determined by a separate validation set, composed of randomly sampled (in terms of $X$) 10% of the data; the MSE on this set for trees with different numbers of leaves is presented on the right in Figure 12 (for the number of clicks outcome) and serves as the final determinant of what we consider the optimal model. The (maximum allowed) number of leaves in the base learner thus serves as our measure of model complexity.

We would like to re-emphasize that the goal here is the inference of the effect of particular parts of the input by way of letting an unrelated, expressive model "test out the competing hypotheses" itself. Specifically, we would like to see such models of adequate but varying complexity *consistently* assign the credit for the outcome to the encodings suggested by the other methods despite not being prohibited from picking up spurious relationships. This differs from other uses of such methods in causal inference, where correctly accounting for unobserved outcomes takes precedence (1; 23).

The figure on the left in Figure 12 shows the inferred *outcome curve* from the model with a maximum of 13 leaves per tree (the best model in terms of the MSE) using the whole of the inputs (including the validation set, which had only been used as a stopping criterion), illustrating that the model is complex enough to be able to fit the observed curve. The training data is included since we want to confirm that the model is able to approximate the generated outcomes sufficiently well, i.e., this is an inference problem.

### A.7.1 THE NUMBER OF CLICKS

The SHAP plot in Figure 13 corroborates the results obtained with the other estimators, picking up bit 1 as the most salient feature in its effect on the number of clicks for each unit, with the same relationship. This confirms that the uncovered relationship between the encoding bit and the outcome is not spurious — i.e., the outcome being just as related to the *incompressible noise $X$* part of the input. Lower values are "bunched" together in a slightly negative effect due to residual entanglement at the lower end of the treatment values. Additionally, the consistency of explanation can be somewhat

observed here by the homogeneity of the coloring — i.e., the lack of dots representing high values in the region of negative impact.

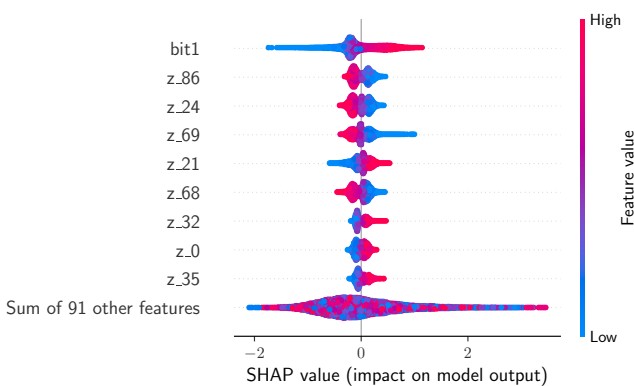

Figure 13: SHAP values for the best fitting model for bit 1.

This consistency of explanation can be better observed in Figure 14, which presents a *heatmap* plot as produced by the `shap` package, with the units being ordered on the $x$-axis in terms of increasing treatment value $t$. The $y$-axis displays the features ordered in terms of their overall importance as measured by SHAP. The plot above the central heatmap plot is the output of the model centered around the explanation's mean value, and the bars on the right display the feature's cumulative contribution.

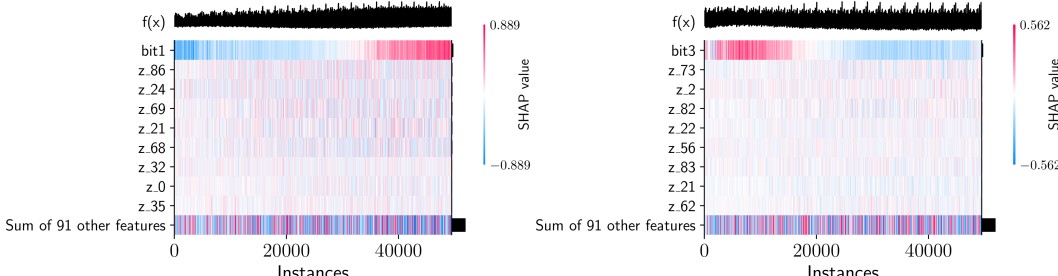

Figure 14: SHAP *heatmap* plot for best-fitting models for bit 1 (**left**) and bit 3 (**right**). The instances are ordered in terms of increasing $t$ from left to right.

The features for individual units are colored in terms of their contribution to the outcome. For bits encoding the observable, we expect to see *local consistency* in the assigned effects since the units are ordered by increasing values of $t$. Conversely, we should observe much less consistently assigned effects for the other bits once the process of disentanglement has reached stationary values. A point we'd like to raise here is that the *CDEV* process does not necessarily start at exactly the same values of treatment for different quantities, so we might not observe full disentanglement for all observables presented. As expected, in Figure 14, we thus observe a greater degree of local consistency in the high-end of the treatment value (as evidenced by the direction and magnitude of the attributed effect for the individual units) for the bit encoding the property — bit 1 — as opposed to bit 3, which has reached the stationary state of disentanglement.

### A.7.2 CLICK SPACING AND REGULARITY

We again apply the same methodology, with the difference being that the observations and corresponding inputs are again stratified by the number of clicks (as in the case of other estimators) and thus regressed separately with boosted tree ensembles of varying complexity. Due to these stratified regressions, it is somewhat more challenging to present consistency across multiple numbers of clicks visually. For the two bits — 1 and 2 — compared in the main paper, we show the corresponding

out-of-sample MSEs for the coda regularity regression as a function of model complexity in Figure 15. The figure demonstrates that more complex models are increasingly finding spurious relationships for bit 2. At the same time, this is not the case for bit 1, for which the preceding estimators suggest encodes the coda regularity. This is another way of saying that the inferred relationship is *consistent* in bit 1 while not in bit 2.

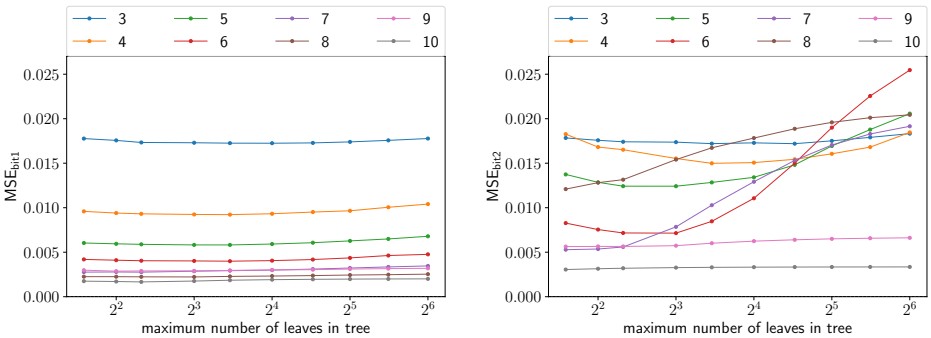

Figure 15: Validation set MSEs for left: bit 1, right: bit 2 for the coda regularity regression stratified by the number of clicks observed.

As in the preceding section, we present the associated SHAP values for the best-fitting models for bits 1 and 2, restricted to codas with *five clicks* due to the above-mentioned presentation issue. We can again make use of the *heatmap* plots to evaluate the degree to which the *disentanglement* process has taken place within the range examined. Figure 16 presents the results for the same two bits as before for the mean ICI, while Figure 17 does the same for coda regularity.

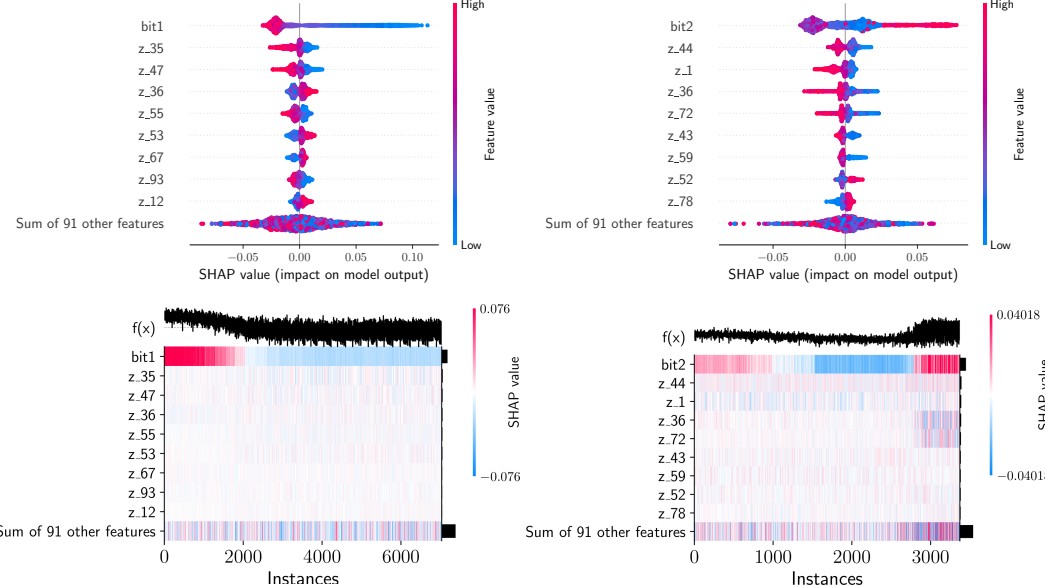

Figure 16: **Coda mean ICI**: **left:** bit 1, **right:** bit 2. **Top:** SHAP values for codas with **five clicks** for the best-fitting models. **Bottom**: heatmap plots show the effects of individual units ordered in terms of increasing $t$ from left to right. Note that the number of outcomes with five clicks differs between the two subsets.

For the mean ICIs shown in Figure 16, we observe the effect suggested by the preceding estimators. We additionally observe greater *consistency* with regard to the expected value in local neighborhoods for bit 1 but not for bit 2, as evidenced by interchanging positive and negative contributions for units with the same treatment value.

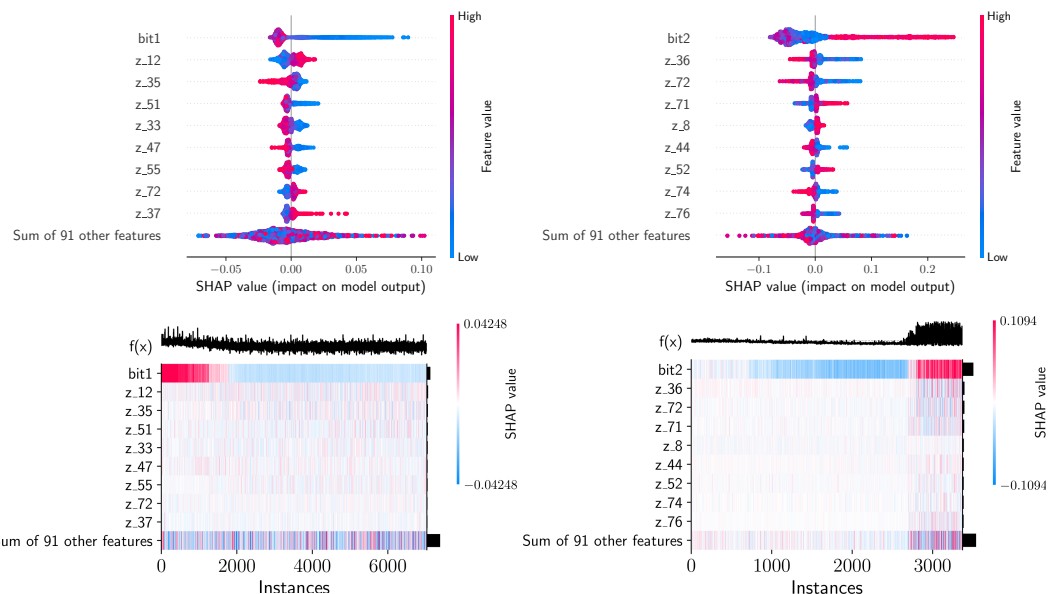

Figure 17: **Coda regularity**: **left:** bit 1, **right:** bit 2. **Top:** SHAP values for codas with five clicks for the best-fitting models. **Bottom:** heatmap plots show the effects of individual units ordered in terms of increasing $t$ from left to right. Note that the number of outcomes with five clicks differs between the two subsets.

Similarly, for coda regularity shown in Figure 17, we again confirm the suggested effect and observe a greater *consistency* of the effect with regard to the expected value in local neighborhoods for bit 1. Overall, the disentanglement process (as seen in the *heatmap* plots) seems to have progressed further for the mean ICIs than for the coda regularity. The *local inconsistency* of the assigned SHAP values in the bits *not* primarily encoding the quantity is a sign of the bit being almost completely disentangled; we can still, however, pronounce the bit as *not* encoding the quantity in question if it (i) displays the observed disentanglement behavior, i.e., moving in unison with other bits, and (ii) is inconsistent across codas with a varying number of clicks, as demonstrated with the other estimators, as well as with varying model complexity, as illustrated in Figure 15.

### A.7.3 ACOUSTIC PROPERTIES

For the click-level spectral mean, we present the results in Figure 18. We again observe the expected patterns when comparing the SHAP values obtained from the best-performing models for the bit that picks up the encoding of spectral mean — bit 0 — and bit 4, which does not. As usual, the inferred effects move in opposite directions with regard to the expected (in terms of SHAP) outcome, with the effects becoming inconsistent in bit 4 once the process of *disentanglement* is complete.

Conversely, the results for the spectral regularity shown in Figure 19 do not display the common inconsistency of assigned effect for values with high treatment in bit 4 — the bit that does not pick up the encoding, meaning that the process of disentanglement has not reached the stationary value yet, illustrating what was suspected when discussing the ATE results in the main paper. Nonetheless, this does not affect the conclusion that bit 3 seems to encode the spectral regularity of a coda since the bit still displays the expected disentanglement behavior indicated by the other estimation approaches.

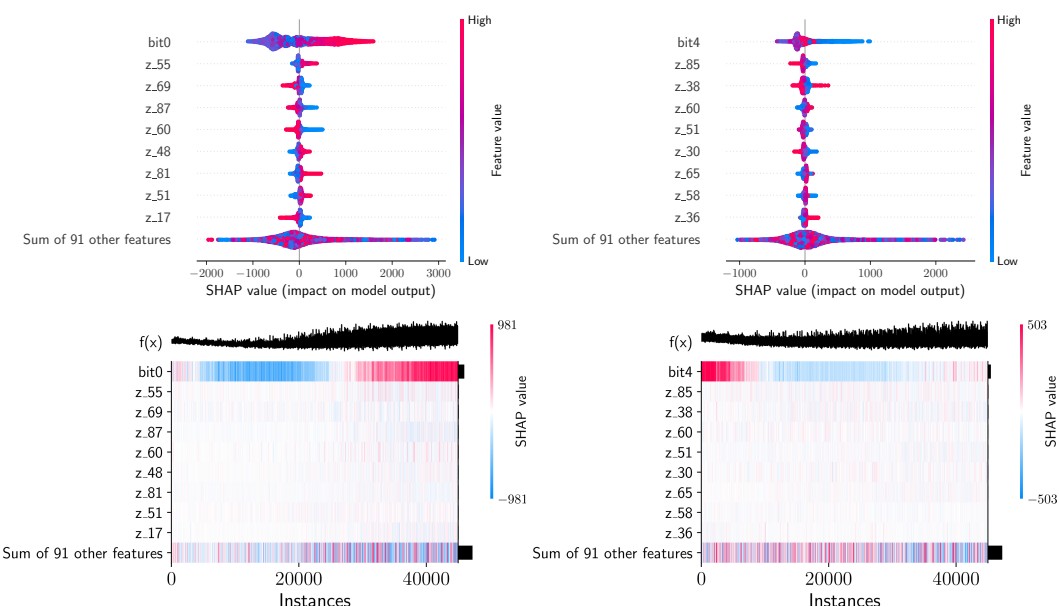

Figure 18: **Spectral mean**: **left:** bit 1, **right:** bit 2. **Top:** comparison in SHAP values. **Bottom:** *heatmap* shows individual units in terms of their increasing value of $t$ from left to right.

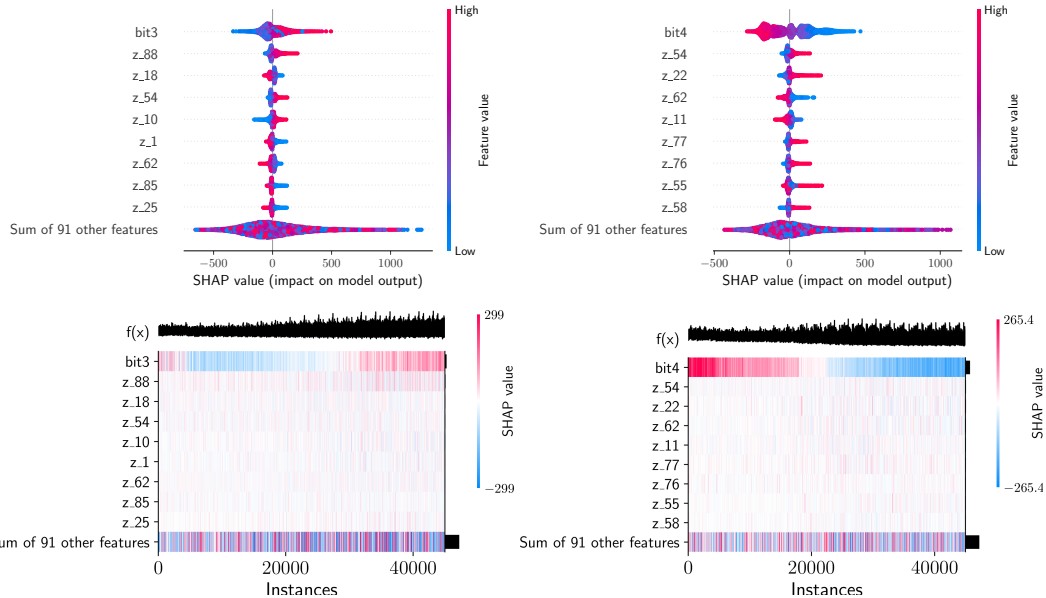

Figure 19: **Spectral regularity**: **left:** bit 1, **right:** bit 2. **Top:** comparison in SHAP values. **Bottom:** *heatmap* shows individual units in terms of their increasing value of $t$ from left to right.

### A.8    A FURTHER DISCUSSION OF THE APPROACH FROM THE VIEW OF CAUSAL INFERENCE

The assumption that we encounter no *fundamental problem of causal inference* (18) itself relies on the assumption that we can "re-set" the unit *characterized by its $X_i$* for each dose $t$. This corresponds to the assumption that $X$ and $t$ have enough informational capacity to encode our observables of interest - in other words, given a specific $X$ and $t$, the observable of interest is *consistent*.

This assumption of consistency of the observables has been borne out in studies with similar datasets: in a work that generated canary bird vocalizations with GANs (28), the authors found that the total

variation of the dataset was sufficiently captured with a latent space length of only 3. Since the training dataset in our case was restricted to the few most popular coda types, we have no reason to severely doubt the consistency assumption, given the total available latent space length of 100. This is also apparent by visually and aurally examining the generated data for fixed $(X, t)$ - the differences, if any, are imperceptible. To summarize - given the whole input $(X, t)$, the observable becomes quite deterministic. On the other hand, only a fraction of $X$ is actually correlated with the outcome of interest (e.g., the number of clicks), and that in a highly convoluted way.

It is thus how we square the seemingly opposing concepts of $X$ being *incompressible noise* only serving as an index to the completely randomized *units* and the fact that conditioning on the whole of $X$ gives us *ignorability*. In other words, the assumption is that the relation of $X$ to the outcome is so complex that sampling it uniformly at random does not produce a noticeable additional effect besides the treatment $t$. This can be likened to the concept of *deterministic chaos*, where a non-linear (in $X$) mapping produces essentially random output, allowing us to perform *completely randomized experiments*. On the other hand, the observables derived from an output generated given a specific $(X, t)$ are consistent, giving us *ignorability* when conditioning on $X$.

A part that remains potentially an issue here for a fully rigorous causal inference argument is that there is less of a guarantee of a *non-interaction* between the treatment and the covariate $X$. Even though the training enforces (via the Q-network and the associated mutual-information loss) $t$ to correspond to consistent output while $X$ can vary without constraints, the process by no means enforces total separation; moreover, what is optimized is actually the mutual information's lower bound (10). This corresponds to potential *treatment effect heterogeneity induced by the covariates*. In cases when the true outcome function is linear, this is not an issue if the covariates $X$ are centered $\bar{X} = 0$, which is the case here. In our case, however, the function is non-linear (and highly complex); therefore, the estimators examined would require that any possible interaction of the covariates $X$ with the treatment $t$ is an approximately odd function (since $X$ are sampled uniformly on $[-1, 1]$) for a more rigorous causal argument.

In contrast, our primary motivation for using causal inference approaches is due to being primarily interested in estimating the *presence of an observed effect* in a very complex system without making additional assumptions or needing access to the internals of the generative process, for which we believe the setup presented is sufficient. Additionally, the (potential) (dis)agreement between the estimators presented can also act as another check on the methodology since they are not direct derivatives of one another.

