# OpenReview forum: "Approaching an unknown communication system by latent space exploration and causal inference"
_ICLR.cc/2024/Conference — Submitted to ICLR 2024_

### Official Review · Reviewer_zdFA · 2023-10-14

**Soundness:** 1 poor
**Presentation:** 2 fair
**Contribution:** 2 fair
**Rating:** 6
**Confidence:** 2

**Summary:**

The authors proposed a method to explore the latent structure of sperm whale's communication system. They combine causal inference and GAN based model into their methodology. They did a thorough analysis of the sperm whale data and proposed some novel insights into the latent structure of the data.

**Strengths:**

The application is quite novel. The paper introduces an approach for exploring scientific findings from a sperm communication system using deep generative models and causal inference techniques.

**Weaknesses:**

The experiments are insufficient. I listed some questions in the section below.

**Questions:**

1. how do the authors know that the latent generated by the proposed method are the actual latent structures of the data not some artifacts of the model? How do you validate your results?
2. there are other disentangement methods using VAE (e.g. betaVAE). The paper lacks comparisons with the existing approaches.
3. in figure 4 and 6, it seems that only 1 bit is different while other bits are in general similar to each other?
4. what are the limitations and future research directions for the proposed method?
5. can the proposed method be generalized to other animals' communication systems or other fields? -- the paper could be more solid by including results on another dataset.

---

> ### Author Response · Authors · 2023-11-19
> **Response to the points raised**
>
> Thank you for the points raised! We will respond in the same order below:
>
> ## How to validate?
>
> Without the ground truth, validation in this sense is impossible by definition. As restated in the general
> remarks, the aim of this work is to provide a method that can guide research by providing independent
> feedback for hypotheses.
>
> That said, the model architecture used as the learning mechanism is structured to encode actually
> meaningful properties of the data, which has been extensively shown on the closest dataset possible where
> ground truth is available - raw human speech. Please note that in that case, the learning was done in exactly
> the same way as here - raw audio, without any domain knowledge provided whatsoever. Furthermore, the
> model is shown to uncover structural relations that are present in whale communication in general, but not
> in the training dataset (Section 4, Paragraph 3). The odds of this happening due to a model artifact are extremely
> low.
>
> ## Comparison with the existing approaches
>
> The relations to other disentangled representation learning methods are discussed in Section 1.2, with $\beta$-VAE specifically being cited as reference number 17.
>
> We feel that the pros and cons of GAN vs VAE-based disentangled representation learning have been discussed in enough detail elsewhere (e.g. reference number 35). What is pertinent to our case is that we have a relatively small training dataset (Section 1.1., Paragraph 2). Additionally, we do not require the posterior latent distribution, nor do we necessarily wish to enforce strict independence between the factors. To illustrate, our model encodes both the mean frequency as well as the acoustic regularity (Section 5), which might not be independent depending on the true ground truth distribution (which we, of course, do not know). However, both properties are salient for e.g. human speech  - eg. general pitch vs. dynamics of a phrase, and it has been shown that whales have the capacity to deliberately control both of them.
>
> ## In figure 4 and 6, it seems that only 1 bit is different
> Disentanglement in this work refers to a process observable in the causal effect, as defined in Section 3: _"which
> lends credence to interpreting this as a process of disentanglement: with higher values, the primary encoded
> effect in each bit starts to dominate, leading the other bits to lose their secondary effects on the number of
> clicks. Thus, we propose that bit 1 primarily encodes the number of clicks"_
>
> As stated in another response: within the training range, the bits remain entangled in their effect; moving
> outside the training range, the primary encoded effect begins to dominate, singling out one bit as the carrier
> of the encoding, while the others return to a baseline value in unison. This phenomenon is replicated across
> different observable quantities and estimators and is further examined in detail via an unrelated method in
> the Appendix, section A.6, which shows that the effect of the bits not encoding the property becomes close
> to random (in any particular observation) once sufficiently disentangled.
>
> Hence the bit that is "different" is the one that disentangles and whose primary effect is the property
> tested, while the others return to a baseline.
>
> ## Limitations, future research directions, and generalizations
>
> The limitations are discussed in the third paragraph of Section 2.
>
> As mentioned in another response and throughout the paper: the overall approach - using a generative
> model to independently encode properties salient to believable data generation, then treating it as a black
> box and uncovering these encodings in a causal experimental setup is completely model-agnostic. There is, therefore, no general impediment to applying the technique to other datasets, be it animal communication or not. In the latter case, one might want to replace the architecture used as the "learner" with another that has been tailored to the nature of the data (and verified on a close dataset with ground truth, as is the case here with human speech). This is one area of future research, with a way to test whether combinations of encodings carry meaning being another area.
>
> In the specific case of animal communication, there is some impediment in the form of the availability of
> good and open datasets, which should be understandable given the quite frequent difficulty in collecting such data.

---

> ### Author Response · Authors · 2023-11-19
> **Follow-up with a new revision**
>
> Dear reviewer,
>
> Thank you for again for the time dedicated to reviewing our paper and the points you raised!  We'd like to let you know that we've updated the paper with a new section (Section A.1.1.) in the Appendix that shows the findings to be robust to the model fit, since they replicate across independently trained models (with different sizes of the encoding space and random initializations).  This adds additional evidence that the effects discovered are not artifacts of the model and should answer your concern expressed in Question 1.
>
> When you had opportunity to read our responses (incl. the general remarks in the overall comments), please let us know if you have any further questions or comments! If you found our responses satisfactory, please consider increasing your score.

---

> > ### Comment · Reviewer_zdFA · 2023-11-21
> > **Response to author's rebuttal**
> >
> > I thank the authors for their detailed responses. They have addressed my questions, and I'm willing to increase my point to 6.

---

> > > ### Author Response · Authors · 2023-11-21
> > >
> > > Dear reviewer,
> > >
> > > Thank you for taking the time to consider the responses and being willing to re-evaluate the paper.

---

### Official Review · Reviewer_H1P3 · 2023-10-18

**Soundness:** 2 fair
**Presentation:** 4 excellent
**Contribution:** 2 fair
**Rating:** 5
**Confidence:** 2

**Summary:**

This paper proposes an approach for uncovering salient properties in an unknown data generating process by leveraging the latent spaces of generative models. Specifically, the authors focus on deepening our understanding of the communication systems of sperm whales using GAN-based generative models. To accomplish this, they propose a method called CDEV which concretely involves setting different latent space variables to extreme values and measuring their effect on observable features using causal inference techniques.

**Strengths:**

Overall, this is a **fascinating paper and well-written paper**. Generally speaking, the notion of using generative models to drive hypothesis formulation and testing for unknown processes may have profound implications for basic science research, and the particular process studied here (whale communication) is refreshingly off the beaten path for ICLR. The proposed CDEV method is explained clearly and with sufficient information to allow for replication (though the data and model appear to be closed).

**Weaknesses:**

I have two significant concerns with the work: (1) that the proposed method is applied too narrowly to be of interest to the broader ICLR community, and (2) that insufficient evidence has been provided to support the application-specific claims.

**Narrow application of proposed methods**. The proposed CDEV method is applied to a single model trained on a small amount of data from a single unknown process. The model is even a fairly unconventional one, with (i) modeling assumptions (the specific use of 5 discrete inputs) rooted in rudimentary understanding of the data process (5 general classes of data), and (ii) an unconventional GAN setup called fiwGAN. The authors claim that “the proposed approach can be extended to other architectures and datasets.” This may be true, although it would certainly be more convincing if the authors demonstrated this themselves. For example, corroboration of the CDEV method on a _known_ process (e.g. human speech) would inspire more confidence. The authors allude to related work on human speech, but while that work does inspire confidence in the generative modeling approach (GANs), it does not look at the CDEV algorithm specifically. Moreover, I have my doubts about the generality of the method, as domain knowledge appears to be a substantive input to the CDEV algorithm at several levels: (i) the model itself, (ii) the data selection process, (iii) the determining of appropriate extreme values, and (iv) the outcome functions used to measure average treatment effect.

**Insufficient evidence to support claims**. Though I am certainly not an expert in this application domain, the authors appear to be making novel claims about the process they study, e.g., that mean spectral frequency is a salient property of sperm whale communication. It is not clear to me that all confounders have been ruled out to support this claim. For example, could it not be the case that fluctuations in mean spectral frequency have nothing to do with _communication_ but are instead caused by some other random source of noise (as an arbitrary example, changes in water temperature)? If this were true, the GAN would still need to produce such fluctuations to fit the data distribution, despite being irrelevant to communication. Another potential confounder: how do we rule out that the GAN simply failed to fit the underlying data distribution, and any interventional effects are simply a consequence of a poorly fit model?

**Questions:**

Other questions and comments:

- Why is the fiwGAN (t, X) structure necessary, as opposed to just training a standard GAN and intervening on the incompressible noise (X)?
- The hypothesis generation seems to be largely expert-driven (i.e., expert codifies hypothesis in outcome function and then tests with model) rather than model-driven (i.e., model conveys some hypothesis that would suggest a particular outcome function). What is the key difference between using a generative model vs. just testing a hypothesis directly on the data distribution (e.g., showing that mean spectral frequency varies across examples in the dataset)?
- How might we extend this method to the particular noise structure of diffusion models, which are more “in vogue” at the moment?
- Figure 1 shows 0, 1 for t codes instead of -1, 1 as in the rest of the paper (this confused me especially w.r.t. choosing low extreme values of -1 rather than 0)

---

> ### Author Response · Authors · 2023-11-16
> **Response to the points raised**
>
> Thank you providing a clear structure to the points raised in your review. For clarity, we will address the points in the same order below.
>
> Regarding the remark about the model being closed: while the original training set is, unfortunately, not yet available, the model, the experiment data, and reproduction code are all provided in full - see Appendix, section A.1 for details
>
> # Weaknesses
>
> ##  Narrow application of proposed methods
>
> ### (i) modeling assumptions ...
> Please refer to the overall comment for clarification regarding the choice of the dimension of t.
>
> ### (ii) an unconventional GAN setup called fiwGAN ... / domain knowledge appears to be a substantive input ... (i) the model itself
> Regarding the generality/conventionality of the model: as outlined in the general remarks, the main reason
> for choosing this setup is that it provides the following: the synthetic data is independently verified, the
> encodings $t$ are consistent and completely unsupervised, and the space available to them is limited, allowing
> us to, together with the preceding two properties, argue for their saliency for believable generation.
>
> The fact that similar models uncover known rules on human speech is used as support of the relevancy of
> the encodings _after_ they’ve been identified via CDEV; in other words, the procedure itself does not require domain knowledge.
>
> CDEV is, indeed, a novel contribution of this work; regarding its generality - the reason for using causal
> inference methods is precisely to establish relationships between the encodings and observable properties
> without additional assumptions or need for external knowledge: e.g., the value of bit 1 _has_ an effect on the
> number of clicks. What that can suggest, however, is reasoned about following the argument in the preceding
> paragraphs.
>
>
> ### (ii) the data selection process,
> The data selection was driven by the fact that whale codas (naturally) come in the context of dialogue, thus
> a significant part of the data featured interleaving codas. Thus, the reason for keeping only the top five
> "classes" is simply that after removing the codas with dialogue, the remaining classes had too few examples
> left - often one or none. As such, this process was not really driven by any goal obtained from domain
> knowledge. Conversely, we posit that a more diverse dataset would be even more beneficial to the generator’s
> ability to uncover salient properties.
>
> ### (iii) the determining of appropriate extreme values
> As outlined in Appendix, Section A.4., the limits of the range of the values used as treatment are determined
> by the quality of output of the generator. This is pointed out as a potential shortcoming in the case that
> these are too narrow, however, there is no domain knowledge involved in that determination - noise is reasonably close to an universal phenomenon.  All values within this range are tested; additionally, the ICE estimator is "location-invariant" as it estimates the effect of an infinitesimal perturbation.
>
> ### (iv) the outcome functions used to measure average treatment effect.
> Please refer to the general comments.
>
> ## Insufficient evidence to support claims
>
> To continue on the points made in the general remarks: the purpose of this work is to present a procedure
> that can _aid_ the scientific process in difficult problems without a ground truth. We do not claim to have
> _proven_ any of the properties identified; however, the agreement of the method with the consensus on the
> more obvious properties, such as the number of clicks, can be used to bolster its credibility for more novel
> suggestions. Keeping this in mind, random sources of noise should even out with enough data, making the
> uncovered encodings more likely to be deliberately controlled by the whales.
>
> Convergence of GANs is, as you’re most likely aware, a complex problem. We’d like to restate that we don’t need perfect convergence
> for this use case; as is illustrated in Figure 3 for an almost worst-case example, the output still looks like a
> whale coda, meaning that the GAN did not fail to fit the data to the extent that would make such analysis
> worthless. As stated in Section 2: _"[noise] might, in part, be due to the network training not having converged
> completely; since the goal is to discover what observable properties are encoded and not high-fidelity whale
> communication generation, this is not a significant impediment."_
>
> In the hypothetical case where one would not be even able to tell when the output is noise (we previously argue that this is hard to believe), one  could simply employ a fail-safe on the overall deviation from the data, for example using something in the vein of the Wasserstein estimators presented in Section 5.

---

> ### Author Response · Authors · 2023-11-16
> **Response to the points raised - Questions**
>
> Continuing from the previous comment:
>
> # Questions
>
> ## Why is the fiwGAN structure necessary?
>
> The reasoning for this particular model setup has been outlined in the general remarks and previous
> points. The incompressible of noise lacks the additional properties outlined of the encoding t, and is
> often "over-parametrized": please see paragraph 2 of Appendix, Section A.8 for an example of a use on
> bird vocalizations, as well as for more details.
>
> ##  The key difference between using a generative model vs. just testing a hypothesis directly...
> Please refer to the general remarks;  as mentioned in another comment - the gist is that we’re interested in what an independent "learner" under the constraint of believability considers salient to a communication system, not whether any given feature is present in the data, which would be trivial for, e.g. the number of clicks.
>
> ## How might we extend this method ...
> The overall approach - using a generative model to independently encode properties salient to believable data generation, then treating it as a black box and uncovering these encodings in an causal experimental setup is completely model-agnostic. The extension to diffusion models is an intriguing problem, about which we're still thinking.
>
> ##  Figure 1 shows 0, 1 for $t$ codes ...
>  Thank you for pointing this out, the typo in Section 2 has been fixed. More explanation is given in the general remarks.

---

> ### Author Response · Authors · 2023-11-19
> **Follow-up with a new revision**
>
> Dear reviewer,
>
> Thank you for again reviewing our paper and your useful comments! We'd like to let you know that we've updated the paper with a new section (Section A.1.1.) in the appendix that shows the findings to be robust to the model fit _and_ the choice of the encoding dimension, since they replicate across independently trained models (with different sizes of the encoding space). This would be unlikely if _"any interventional effects are simply a consequence of a poorly fit model?"_
>
> When you had opportunity to read our responses (incl. the general remarks in the overall comments), please let us know if you have any further questions or comments! If you found our responses satisfactory, please consider increasing your score.

---

> > ### Author Response · Authors · 2023-11-22
> >
> > Dear reviewer,
> >
> > We'd like to kindly remind you to examine our responses and follow up if any further clarification is needed. If you found our responses satisfactory, please consider increasing your score.

---

> > > ### Comment · Reviewer_H1P3 · 2023-12-05
> > >
> > > My sincere apologies for the extremely delayed response. I am satisfied with the authors' response to my concerns around the unorthodox GAN training setup. However, I remain concerned that the proposed method is applied too narrowly here and am unconvinced regarding its applicability to other tasks or interest to the broader ICLR community. Accordingly, I will raise my score from reject to weak reject.

---

### Official Review · Reviewer_w8kr · 2023-10-31

**Soundness:** 2 fair
**Presentation:** 2 fair
**Contribution:** 2 fair
**Rating:** 3
**Confidence:** 4

**Summary:**

This work applies an existing approach called fiwGAN to a “sperm whale communication” dataset in order to extract an interpretable representation. They then assess whether the learned code “aligns” with natural features of whale signals such as “number of clicks”, “clicks regularity” and “click spacing” by perturbing the components of the latent code individually to see if they encode these natural features.

**Review summary:** Although I was glad to see an application of disentangled representation learning to real world data, I found this work unconvincing. I have issues with the motivation, the originality, their use of the term "disentanglement", the use of causal terminology and clarity. I also found the experiments to be unconvincing. I expand on these below. For these reasons, I recommend rejection.

**Strengths:**

- Original application to whale communication.
- Surprising findings, in Figure 1: “The 9R type was never part of the training data, yet our network learns to generate codas that resemble this type (Sec. 4).”

**Weaknesses:**

### **Unclear motivation**
I have a general concern. What is the advantage of learning a representation (t,x) using a deep generative model in the hope that $t$ will represent some intuitive features (like number of clicks and click regularity), if these very features can be extracted from the raw data algorithmically (as is done in this work, for evaluation)? I understand that this is only to evaluate the disentanglement of the method, but if these intuitive features can be computed from the raw data, what’s the point of trying to extract them in an unsupervised manner here in this application?

### **Originality**
As I said above, I believe this work present an original application of disentangled representation learning. However, I don't see novel methodological contributions. The authors essentially trained an existing method and applied fairly standard qualitative evaluation strategies to assess disentanglement.

### **Unclear use of the word “disentanglement”**
- The authors should clarify what they mean by disentanglement, since their definition (which I inferred from their experiment) seems different from the one I know and which is widely used: Typically, disentanglement means the following: every component of the latent representation influences one and only one interpretable factor of variation.
- Figure 4: This does not correspond to the usual notion of “disentanglement”. Here, all the dimensions of t have an impact on the number of clicks. The interesting thing is that it seems only one dimension of $t$ has a positive impact, while all the others have a negative impact.
- Figure 5: I still believe this does not correspond to the usual notion of disentanglement, since, again, every dimension of t_i have an impact on the “inter-click” interval and the “coda regularity”, it’s just that one of the bits has a monotonic influence while the others are non-monotonic.

### **Concerns with experiments**
The experiments did not convinced me that the fiwGAN algorithm can robustly discover natural factors of variations in the data:
- Are the findings robust to reinitialization? I.e., if you rerun this experiment with multiple seeds, are you always finding similar patterns? I don’t think the experiments show reruns. This is very important for such analysis, to avoid cherry-picking.
- In Figure 4: the curves that go down seem to go up again after t = 10, do they end up being positive if we extend the range of t even further?
- Surprising findings, in Figure 1: “The 9R type was never part of the training data, yet our network learns to generate codas that resemble this type (Sec. 4).” I would like to see stronger evidence that the model can generate reasonable samples never seen during training. Was this sample cherry-picked?

### **Unnecessary use of causal terminology**
The framing in terms of causal inference feels a bit unjustified, since one has full control over both x and t, so that there is not direct causal effect from x to t, making causal inference trivial.

### **The work lacks clarity at times**
- In intro: “Networks trained on raw speech data are shown to learn to associate lexical items with code values and sublexical structure with individual bits in $t$ …” the variable $t$ was not defined, so it’s hard to follow what is meant here.
- I found the description of fiwGAN a bit imprecise at the beginning of Section 2. I think it would help to write down explicitly the loss that is optimized, even if this is not part of your contribution.
- Figure 2: The code t is binary, but the rest of the paper assumes t is Rademacher, i.e. with support {-1, 1} instead of {0, 1}. Which one is it?
- Section 3: “We observe a high degree of entanglement (4) of the learned encodings within the range seen in training ([−1, 1]).” I don’t understand the meaning of “entanglement” here. I now understand that this paragraph refers to experiments in Figure 4. I feel like the separation between theory/method and experiments is not clear enough here.

**Questions:**

- $t$ is a vector of 5 bits, to fit the number of coda types in the dataset. Is this for the approach to work? You say other works have shown that having a mismatch doesn’t matter. But does it matter in your specific application?

---

> ### Author Response · Authors · 2023-11-16
> **Overall answers to the points raised**
>
> Thank you for a well-structured response; below, we'll respond in the same order of points raised.
>
> # Weaknesses
>
>
> ## Unclear motivation
>
> Please refer to the general remarks for a more complete restatement. The gist is that we’re interested in
> what an independent "learner" under the constraint of believability considers salient to a communication
> system, not whether any given feature is present in the data, which would be trivial for, e.g. the number of
> clicks. The disentanglement with CDEV is the method of uncovering these encodings.
>
>
> ## Originality
>
> We believe this is the first example of using causal inference methods for interpreting encodings in an
> unsupervised setting (please refer to Section 1.2 for more discussion on related work). We also believe that
> the method of disentanglement - CDEV - is also novel: we could find no prior work that forced extreme
> values on the input and measured the effect systematically with causal inference methods. We see no need for
> additionally inventing separate causal inference estimators since using proven ones adds credibility to the
> results obtained via CDEV. That said, we do believe that the ICE estimator might not be that well known
> and is a very good fit for this (and similar) use cases, which could be of broader interest.
>
>
> ## Unclear use of the word “disentanglement”
>
> The definition is given in Section 3: _"We observe a high degree of entanglement of the learned encodings
> within the range seen in training ([1, 1]). However, when setting the treatment value above that range, all the
> bits but bit 1 stabilize at roughly a constant effect, which lends credence to interpreting this as a process of
> disentanglement: with higher values, the primary encoded effect in each bit starts to dominate, leading the
> other bits to lose their secondary effects on the number of clicks"._
>
> While we agree this definition might be relatively unusual, it was inspired by the phenomenon of Quantum
> (dis)entanglement, in which case the use of the phrase is well-established. In this view: within the training range,
> the bits remain entangled in their effect; moving outside the training range, the primary encoded effect begins
> to dominate, singling out one bit as the carrier of the encoding, while the others return to a baseline value in
> unison. This phenomenon is replicated across different observable quantities and estimators and is further
> examined in detail via an unrelated method in the Appendix, section A.6, which shows that the effect of
> the bits not encoding the property becomes close to random (in any particular observation) once sufficiently
> disentangled.
>
> We thus think the name is still a good descriptor of the phenomenon observed; however, we are naturally amenable to renaming the method
> if the confusion is believed to be too great.
>
> Regarding Figure 5, the plots presented are for a single bit, stratified by the number of clicks. The bit proposed has a consistent
> effect no matter the number of clicks in the generated coda, while other bits (such as the one presented) do not present any consistency in their encoding. This is in line with the definition presented above.
>
> ## Concerns with experiments
> ### Re-initialization
> The results of re-initialization are presented in Figure 8 in the Appendix, Section A.5.1, showing the results
> to be very robust to re-initialization.
>
> ### Figure 4
> As mentioned in Section 2 and the Appendix, Section A.2., the upper range was selected on the basis of
> the generator still outputting samples with low enough noise so that our algorithmic measurement of the
> observables was possible. To answer the question - extending the range would give us data that makes no
> sense, hence speaking of an effect above (roughly) that value is not possible.
>
> ### Surprising findings
> The evidence can be inferred from the relationships displayed in Figure 5. We see the presence of codas with
> click numbers up to 10, while the model infers the connection between the number of clicks and the regularity
> on its own: _"Furthermore, [real] codas with more clicks are often more regular in their ICIs, regardless of
> their duration. The generator has thus inferred this connection without the codas with a high number of clicks
> even being present in the dataset."_
>
> Thus, regular 9R-like (the R denoting "regular") codas are quite common in the generated samples.
>
> ###  Unnecessary use of causal terminology
>
> While this is true to an extent (and argued for in the Appendix, Section A.8), we believe the work shows that
> causal inference-derived estimators are a natural fit for this problem setup. This is the reason phrases like
> "methods from causal inference" are used in the work instead of a full causal argument. As an example of the
> fit, the ICE estimator neatly sidesteps most dilemmas with regard to the choice of the baseline and the range
> of "treatment", which are not self-evident in this case.

---

> ### Author Response · Authors · 2023-11-16
> **Overall answers to the points raised - continued**
>
> # Weaknesses
>
> ## The work lacks clarity at times
>
> ###  variable $t$ was not defined
> Thank you for pointing this out, this has been fixed.
>
> ###  it would help to write down explicitly the loss that is optimized
> The exact loss has been added in the Appendix, Section A.2.
>
> ###  The code t is binary, but the rest of the paper assumes t is Rademacher
> Thank you for pointing this out, this has been fixed - see general remarks.
>
> ###  the meaning of "entanglement"
> Please refer to the discussion regarding the use of the phrase "entanglement" in the previous comment.
>
>
> # Questions
> ###  Do we need 5 bits for the approach to work?
> Please refer to the general remarks.

---

> ### Author Response · Authors · 2023-11-19
> **Follow-up with a new paper revision**
>
> Dear reviewer,
>
> Thank you for again reviewing our paper and your useful comments! We'd like to let you know that we've updated the paper to further address two of the points you raised, namely robustness to re-initialization, as well as robustness to the size of the encoding space. The results can be found in the Appendix, Section A.1.1. and should answer the questions in the affirmative: the approach is robust to both.
>
> When you had opportunity to read our responses, please let us know if you have any further questions or comments. If you found our responses satisfactory, please consider increasing your score.

---

> > ### Comment · Reviewer_w8kr · 2023-11-22
> >
> > Thanks for considering my comments seriously and addressing them.
> >
> > Motivation: I'm still a bit confused about the motivation. I understand that this work is about representation learning. My understanding is that it is concerned with learning a generator which takes as input a code t that is interpretable, correct? Here you demonstrate that your method achieves this, in the sense that there is a correspondence between the dimensions of t and some observables (like number of clicks, for example). My concern remains: I don't understand the usefulness of doing this. Let me try to explain my confusion by explaining a situation where I believe learning a disentangled representation makes sense: Observations are images of a scene with various attributes like color of object, color of background and camera angle. In that case, learning a disentangled representation means learning a encoder function which, when applied to an image, outputs those attributes like color etc... (up to permutation and element-wise rescaling). The nice thing about this is that we don't know of another way to extract these interpretable features, i.e. learning is necessary. My issue with this work is that, the interpretable features in this application, like number of clicks ... etc. can be extracted from the signal with a simple algorithm, i.e. no need to learn an encoder to output those. Thus my question is this: why should we bother learning an encoder in this application, given that we know how to extract the interpretable features without having to learn anything? I'm sorry if I'm very confused, but that's my current understanding.
> >
> > Novelty: My understanding is that the idea of setting latent codes to extreme values was already explored by [1]. Are you saying that the methodological novelty lies in the use of causal inference? If so, I remain unconvinced, since the causal inference aspect of this work appears trivial to me (see later point about causality terminology). Could you contrast further your approach with the methodology of [1]?
> >
> > Regarding the use of the word "Disentanglement": I think it's ok to use that word differently from the literature I know. But I think you should be careful spending more time defining what you mean here, and maybe contrast a bit more with other known definitions.
> >
> > Robustness to re-initialization: Showing one rerun with a different reinitialization is not enough in my opinion. How about 10 reruns? This is rather standard in many works.
> >
> > Causal terminology: I read Appendix A.8 and I stand by my comment. The whole point of causal inference is to deal with confounding. But here there's no confounding by design, since X does not cause t.
> >
> > My main concerns remain unaddressed, so I have to keep my score unchanged. If my answer still shows important misunderstandings, I'm open to have them corrected.
> >
> > [1] G. Beguš. Generative adversarial phonology: Modeling unsupervised phonetic and phonological
> > learning with neural networks.

---

> ### Author Response · Authors · 2023-11-22
> **A second response (1)**
>
> Thank you for providing more details regarding your concerns; we've striven to answer them as completely as possible given the additional information.
>
>  # Motivation
>
> Restating the clarification of our motivation from the general comment:  we’re interested in what an independent learner under the constraint of believability considers most salient to an unknown communication system. K.P. Murphy's Probabilistic Machine Learning: Advanced Topics describes it thus in Section (32.3.2.1): _"One criterion for learning a good representation of the world is that it is useful for synthesizing observed data. If we can build a model that can create new observations and has a simple set of latent variables, then hopefully this model will learn variables that are related to the underlying physical process that created the observations."_
>
> To answer the question directly regarding extracting the features from the data directly: we are naturally biased towards assigning meaning to features that appear meaningful **to us** in truly unknown data such as sperm whale communication. For example, we can count the number of clicks in a vocalization "by hand". Can we then simply pronounce this as a carrier of meaning? How are we sure it's not simply a side-effect of something else? To provide an analogy, surely bats hear "features" of our speech that carry absolutely no meaning to us since we cannot hear those frequencies - they are merely high-frequency artefacts.
>
> As outlined in the general comment and the previous responses, the architecture gives us a way of letting an **independent, unbiased** learner suggest what it considers most salient.
> The learner has the following properties: (1) it has to be believable to an independent observer with access to real data, (2) the special encoding $t$ has to be consistent (3) it has to be salient else the model is wasting a precious encoding spot.
>
> Thus, to answer your first question, this is in essence *not* about ensuring that $t$ is interpretable, us having already decided what it should be. We aim to answer **what** gets encoded in an unsupervised learning, no ground truth situation. Therefore, we believe there has been a misunderstanding about the basic aim of the paper.
>
> # Novelty & Causal terminology
>
> As a reminder, for GANs, the "incompressible noise" $X$ also encodes information, that is, its values have direct effects on the outcome. Thus, "ordinary" GANs have just $X$ as their feature space. However, it does not have the above-mentioned properties (2) & (3). Thus, the goal here is to identify **what** *gets encoded in $t$ as opposed to merely $X$*.
>
> The novelty compared with [1] is described in paragraph 2 of Section 2. To expand on that: [1] is fundamentally concerned with examining how neural networks learn features of data with ground truth - human speech. The forcing of extreme values is employed in a latent space _interpolation_ problem setting (cf. chapter 20.3.5 in KP Murphy's book cited previously), with the architecture being an "ordinary" GAN. Thus specific, **pre-determined**  phonetic features are focused on. Extreme values are employed in the context of manipulating single bits in $X$ while the rest are set at _constant_ (random) values _for a given, single, example in which the desired phonetic property has already been identified_. This enables the author to examine, for example, how that specific phonetic feature morphs when one traverses the latent space outside the bounds of the training range. While the regression (not testing) methods employed are appropriate for an interpolation setting, there is no experimental setup with large samples of experimental _units_ being tested.
>
> Conversely, this work is in almost all aspects an inverse problem to the above. We are dealing with a _structure discovery_ problem (Chapter 20.3.4 in the same book), where we have no ground truth. Thus we situate our methodology within an _experimental setup_ and wish to **test** for **what** properties are encoded. The **criterion** of something being encoded in a bit of $t$ is it undergoing the _process of disentanglement_ in the causal effect space with sufficiently large "treatment" values in the same bit. The extreme values technique is used here (in contrast to [1]) to **force** this process of disentanglement, which is not studied in [1].

---

> > ### Author Response · Authors · 2023-11-22
> > **A second response (2)**
> >
> > (continuing prev. section)
> >
> > ## Use of the causal setting
> >
> > Since $X$ have an effect on the observables tested (cf. first paragraph of this section), we need to rule that out if we wish to determine the effect of the encoding in $t$, which has the special properties outlined multiple times before. To answer your question: the issue, therefore, is not the confounding effect of $X$ on $t$, but the effect of $X$ on the outcome. Therefore, this is ruled out by setting the examination up as a _completely randomized experiment_ (CRE).
> >
> > Having arrived at this point, we still need _something_ to identify an effect that points out the encoding. Using causal inference estimators is the most natural choice given this setup. To answer the question directly, while confounding necessitates the usage of causal inference estimators, being able to set up a completely randomized experiment _in no way_ precludes one from using them to estimate the effect - the ATE estimator is _still_ the basic estimator here. Furthermore, the architecture setup with separate $t$ and $X$ maps almost perfectly to continuous treatment causal inference. As explained in Section 2.1, the only point of contention is the lack of the baseline for the "dose", which is resolved elegantly by the ICE estimator.
> >
> > In summary, causal inference estimators give us the perfect match for _quantifying_ the process of disentanglement forced by the extreme values, that is, to identify encodings. This, we call the CDEV method.
> >
> >
> > ## Summary
> >
> > Thus, we believe that the methodological novelty of this work lies in the complete methodological *approach* that is first presented here. This approach consists of the (1) architectural setup (with the properties described in the motivation section) that facilitates (2) the experimental setup that is analyzed using the (3) CDEV method in order to provide novel guidance for answering questions in a (4) quite novel problem setting.
> >
> > # Robustness to re-initialization
> >
> > We're not entirely clear what specific aspect is meant by "re-run" here. Below, we assume that the argument is that there is a significant probability of the results presented being due largely to chance. We'll address all interpretations that come to mind:
> >
> > - **re-run of experiments given a trained model**:
> > Appendix A.1.2. shows _almost identical_ effect values for independent experiments, each with a differently randomly sampled (i.e. different random seeds) samples of 2500 units. It is almost a tautology that if two random sequences converge to the same value, the limit must exist. To put it in a different way: the odds of a statistical estimator giving two nearly identical effect curves with two large, independent samples, _yet_ the true effect curve not existing are extremely unlikely.
> >
> > - **re-training of the model**:
> > Appendix A.1.1 shows that the same property is encoded when a separate model with a different encoding space size is trained, with the same _process of disentanglement_.
> > Since GANs are trained via a minimax game, which almost surely has no equilibrium in cases such as here where we've added loss terms, the odds of two such games for different models  (1) converging to good output generation (2) encoding the same property in a completely unsupervised fashion  with (3) the same disentanglement process _purely by chance_ are, again, astronomically unlikely. This is the flip-side of the difficulty of training GANs.
> >
> > - **replicability of the  disentanglement process:**
> >  The same disentanglement process, which is used to identify encodings, is observed across completely unrelated properties and estimators, including a non-causal inference method presented in the Appendix, section A.7.

---

> > > ### Comment · Reviewer_w8kr · 2023-11-22
> > >
> > > I thank the authors once again for bringing clarifications about their work.
> > >
> > > Motivation: I'm not sure whether it is fair to conclude that, because the fiwGAN model extract some features, it means that these features are important for communication and not an artefact of the signal. This reasoning seems flawed to me. I don't see any reason why the model couldn't encode artefacts (i.e. features that are not relevant to communication)
> > >
> > > Causal terminology: I respectfully disagree with the authors. I think introducing the jargon of causal inference here is "overkill" and does not serve the reader. What has been done here are standard latent traversals (with additional averaging over $X$). Sure you can always explain these in the language of causality, but I don't think this brings something new to the table.
> > >
> > > Robustness: I was talking about rerunning the learning phase multiple times. I saw that you added an extra rerun in the appendix A.1.1 showing the same results. My point is that this is only two runs, I think this is not enough. Could you show 10 runs? What fraction of these 10 runs shows the desired behavior?
> > >
> > > I'm sincerely sorry, but I don't feel comfortable recommending this paper for acceptance. My suggestion for a next iteration of the work would be this: Clarify the motivation further in the paper, lighten the use of causal language since IMO this is a trivial application of causal inference (if you want you can mention on the side that these estimators correspond to causal estimators), and provide more reruns of the model training with different initializations to show that the findings are robust (this is fairly standard in the disentanglement literature).

---

> ### Author Response · Authors · 2023-11-22
> **A clarification**
>
> Thank you for again singling out the points of disagreement; unfortunately, we still feel there has been a misunderstanding of the paper on some of them.
>
> **Motivation**:
> _I'm not sure whether it is fair to conclude that, because the fiwGAN model extract some features, it means that these features are important for communication and not an artefact of the signal. This reasoning seems flawed to me. I don't see any reason why the model couldn't encode artefacts (i.e. features that are not relevant to communication)_
>
> We have _never_ claimed that what the model encodes is guaranteed to be meaningful. The aim of this application is to provide a method of obtaining a **second opinion** that is derived independently and does not share human biases, yet it has to be believable (+ the other properties listed previously). This is spelled out in the introduction of the paper and has been re-iterated in the rebuttals. This second opinion can then be used, for example, by researchers to help decide which research direction to prioritize.
>
> That said, we have very good empirical reasons to give credence to the model. As stated in the first general comment:
> "Additionally, the fiwGAN architecture has been shown to uncover linguistic rules on human speech (references 3; 5; 2), hence we suggest (but not claim) that the encodings uncovered here might play a similar role in whale communication. This is further bolstered by the fact that it independently learns a rule present in real-world codas, but not in its training data (Section 4), as well as determining that the number of clicks is informative, which is the basic consensus [among the experts] in the domain."
>
> **Causal terminology:**
> _I respectfully disagree with the authors. I think introducing the jargon of causal inference here is "overkill" and does not serve the reader. What has been done here are standard latent traversals (with additional averaging over $X$ ). Sure you can always explain these in the language of causality, but I don't think this brings something new to the table._
>
> It is **precisely** the ruling out of the effect of $X$ that is needed for identification of the effect of $t$. Thus, the "additional averaging over $X$", which is what this amounts to when (and only when) using the ATE estimator (one of three approaches presented) is a **crucial** feature. A standard latent traversal does not work here since you cannot proclaim the effect to be encoded in the "special" $t$ space without ruling out the possibility of it being an artefact of $X$. Any restatement of these requirements in other language would, in practice, amount to re-inventing causal inference for randomized experiments, making it less clear to the reader.
>
> **Robustness:**
> _I was talking about rerunning the learning phase multiple times. I saw that you added an extra rerun in the appendix A.1.1 showing the same results. My point is that this is only two runs, I think this is not enough. Could you show 10 runs? What fraction of these 10 runs shows the desired behavior?_
>
> This should be clarified in light of the misunderstanding regarding the motivation of the paper discussed above. We aim to present a _methodology_ of extracting a "second opinion" from independent learners that don't share human biases.
> The methodology replicates perfectly, since a virtually identical disentanglement phenomenon can be observed and quantified after using the CDEV method on an independent model.
>
> We feel that the requirement for an ensemble of 10 (or 100) of independent learners to agree 100% on all of the salient features is overkill and missing the point of the paper. Even in that case, we would _still_ not proclaim these to be the true carriers of meaning of in whale communication, as explained in the motivation rebuttal above.

---

### Official Review · Reviewer_iB7G · 2023-11-01

**Soundness:** 3 good
**Presentation:** 4 excellent
**Contribution:** 3 good
**Rating:** 8
**Confidence:** 3

**Summary:**

This paper reports a very fascinating application of the use of machine learning and data mining for scientific knowledge discovery, using the example of understanding the communication systems of non-human organisms.
Specifically, the authors extract the latent structure of the target data based on the fiwGAN architecture, a state-of-the-art method of the WaveGAN model's genealogy of methods without information loss (e.g., loss of phase information when turned into a power spectrogram), manipulate individual units of the network inputs to extreme values and The proposed approach is to manipulate individual units of the network's inputs to extreme values and estimate their impact on the observable properties of the outputs using causal inference methods.
Section 2 describes the latent structure learning method. The paper then investigates the latent causal structure of three human-interpretable acoustic features. Following this, Section 3 focuses on the number of clicks, Section 4 on the click interval and its standard deviation, and Section 5 on the timbre (spectrum) itself and its variation (standard deviation).
Finally, the paper summarizes new scientific findings derived from machine learning in Section 6.

**Strengths:**

In response to the challenging task of unraveling the communication system of whales, this paper proposes a new causal analysis method for state-of-the-art machine learning methods for acoustic signal analysis, inspired by methods for inferring dose-dependent responses in the context of pharmacology. The strengths I perceive in this paper can be summarized as follows.
- The subject matter itself that this paper deals with is of great scientific importance beyond the toy example.
- The presentation of this paper is very excellent. It is well thought out for a diverse audience, especially since it provides sufficient background knowledge and motivation for the research questions, even for the acoustic communication systems of whales, for which many machine learning researchers may not have domain knowledge.
- The methods proposed to gain insight into the causal structure of the target of interest from the latent variable space extracted by machine learning data analysis are very appealing, a new approach to cross-disciplinary thinking inspired by pharmacology dose-response.

**Weaknesses:**

I have not found any notable weaknesses in this paper. If I had to pick one, I would say that I do not have domain knowledge of how impactful the new findings of this paper on the challenge of elucidating the communication system of whales (the finding that not only clicks but also tones themselves may have important hidden meanings) are in the area of expertise in question.

**Questions:**

First of all, I would like to thank the authors for sharing this paper with the community. I have enjoyed reading this paper very much. To make sure I understand the value of this paper correctly, let me ask the authors two questions.

(1) The contribution of the explanation to the original data of the latent dimension specified in the 5 dimensions being extracted by fiwGAN.
My question is: Is it known how much the ATE of the number of clicks contributes to the representation of the observed data?
More specifically, does the fact that the bit with index 1 in the latent variable space (5 dimensions) responds well to ATE mean that ATE is the second principal component (in the sense of, say, principal component analysis)?
If so, I would be very interested in what elements are responding to the first principal component (i.e., the bit with index 0).
I understand the part where the model is limited to 5 bits (32 classes) of characteristic coding space for the 5 coder types present in the data so that constructivity can be captured. Is the index of these bits tied to foresight knowledge such as the index of these 5 coder types? Or does the index of bits reflect some ranking of its expressive power, as in, for example, classical principal component partial analysis?
This may be a simple question that arises because I simply missed the description of the details.

(2) Does the method of identifying causal structures analogous to dose-response, which the current proposed method does, also work for factor combinations?
My understanding is that the current causal structure is investigated separately for each factor, such as number of clicks, tones, etc. Is it not possible, for example, for a combination of those factors to have a new semantic meaning?
Is it possible, for example, that "a constant rhythm of low tones" and "a constant rhythm of high tones" have different meanings? I imagine that it would be difficult to distinguish between the two with only one factor, "constant rhythm," and that it might be difficult to tell without combining factors.

---

> ### Author Response · Authors · 2023-11-18
> **Response to the questions raised**
>
> Thank you for raising some interesting points! We'll respond to them in order below:
>
> ## Order of bit indices
>
> As mentioned in Section 2, and the newly added Section A.2 of the Appendix, the encoding is enforced to
> be consistent in a loose way, via an additional optimization loss - the variational lower bound between the
> generated data and the encoding value. Hence, we cannot reliably claim the rank ordering of the bits to
> indicate the properties’ importance in the vein of say, PCA. The focus of the approach, therefore, is more to identify (via causal estimators) _what_ is encoded.
>
> ## Factor combinations
>
> Factor combinations, or "bit interactions", is indeed an area we’re examining for the extension of this method, in addition to adopting it to other architectures. As your motivating example shows, the presence of these is eminently plausible; additionally, the specific architecture has been shown to learn with compositionality on human speech, where the ground truth is available.

---

> > ### Author Response · Authors · 2023-11-19
> > **Follow-up with a new revision**
> >
> > Dear reviewer,
> >
> > To follow up on your interesting question (1): we've added a section in the Appendix (A1.1.) that demonstrates our answer in the previous comment: in a new model, the number of clicks is encoded in bit0, while in the main model presented it was in bit1.
> >
> > Please let us know if you have any further questions or comments!

---

> > > ### Comment · Reviewer_iB7G · 2023-11-23
> > > **Thank you for kind response and revised paper.**
> > >
> > > I greatly appreciate the author's thoughtful responses and insightful discussions with the other reviewers.
> > > My two concerns have been addressed. I am very grateful to you for adding explanations related to them in the revised paper.
> > > I again find the examples of applications of machine learning to real science that this paper provides very important. I would like to keep my grade intact.

---

### Author Response · Authors · 2023-11-16
**General remarks and changes to the latest revision**

First of all, we'd like to thank you for the effort and time you dedicated to this work!
We have uploaded a new revision of the document (but not the supplement), which contains the following minor changes:
- Section 2 erroneously mentioned the encoding being sampled as Rademacher variables instead of Bernoulli.
- Details regarding the losses optimized by the architecture have been added to the supplement in Section A.2
- mention of the encoding variable $t$ before definition in the Introduction has been removed

In the following sections, we'd like to answer some common questions.


# Overall idea behind the work

For clarity, we’d like to briefly re-state the general idea behind the proposed setup. The overall motivation is
captured in the Introduction: _"If sperm whales encode information into their vocalizations and our model
can learn to imitate those well, the encoding in our models can likely reveal what might be meaningful in the
sperm whale communication system."_. In other words, we’re interested in what an independent "learner"
under the constraint of believability considers most salient to an unknown communication system.
This is in line with the motivation provided in the relevant chapter (32.3.2.1) in K.P. Murphy's Probabilistic Machine Learning: Advanced Topics for generative representation learning in general:  _"One criterion for learning a good representation of the world is that it is useful for synthesizing observed data. If we can build a model that can create new observations, and has a simple set of latent variables, then hopefully this model will learn variables that are related to the underlying physical process that created the observations."_


The way this is achieved is to take advantage of the architecture and causal inference methods: first,
since the learning mechanism is a GAN, the generator is independently checked by the discriminator with
access to real data. Second, the encoding t is ensured to be consistent by the additional mutual information
loss between the Generator and the Q-Network (Section 2, paragraph 1). In other words, the generator is
incentivized to encode observable (since the Q Network only receives the generated data) properties into the
t space, but is free to determine which ones without supervision, subject to the need for the synthetic data to
be believable to the discriminator. Third, by the fact that the space of t is limited in size, these are likely to
be salient. Fourth, the identity of the encodings is then uncovered in a randomized experiment setup with
the help of causal inference estimators.

Additionally, the fiwGAN architecture has been shown to uncover linguistic rules on human speech
(references 3; 5; 2), hence we suggest (but not claim) that the encodings uncovered here might play a similar
role in whale communication. This is further bolstered by the fact that it independently learns a rule present in real-world codas, but not in its training data (Section 4),  as well as determining that the number of clicks is informative, which is the basic consensus in the domain.

Overall, the proposed use of this setup is to help guide research by providing independent feedback for
hypotheses. In so far as it is a weakness, the need for the latter to be provided is spelled out quite clearly in
the paper. Nothing, however, precludes one from generating these automatically, if that is possible given the
specific problem examined.


# Choice of the encoding space size
Regarding the choice of 5 as the number of bits in the encoding space t: while this is a free parameter, the
particular choice is only a very rough guide: if the marine biologists believe there to be 5 classes in the
dataset, we’d naturally want the setup to be able to encode five separate features, which indicates 5 as a
good choice. However, as illustrated in Section 2, paragraph 2, mis-specifying this number has little effect on
the system’s ability to encode meaningful properties, which is what we mainly want it to do (i.e., this is not a clustering problem).


# $t$ distribution during training, limits during the experiments
The typo in Section 2 has been fixed: the training encoding is sampled as Bernoulli variables, not Rademacher
(the model code is included in the supplement, please refer to the Readme.md file within).

The limits of the "treatment" range t tested, however, are driven by the quality of output of the generator - please refer to Appendix A.4. When the phrase "training range" is used, this includes the range of the incompressible noise, hence [−1, 1]. As a side note: moving the lower range of t tested from -1 to 0 would _strengthen_ all the results presented.

# Reproducibility
While the training data is, unforunately, not available, the model, the model code, the experiments code, and intermediate results (on the level of the observables) are all provided. Please refer to the Appendix, section A.1, and the README.md file in the accompanying supplement.

---

### Author Response · Authors · 2023-11-19
**New revision - robustness to re-initialization and choice of encoding space size**

Dear reviewers,

We'd like to again thank you for reviewing our paper and your useful comments! In light of these, we've added a new section - A.1.1 - in the appendix that demonstrates the same findings on an independently trained model with a different choice of encoding space size.
This shows that the method and findings are robust both to the artifacts of model training (re-initialization) _and_ the choice of the number of desired encodings in $t$.

The link to the additional model is included in the Appendix, Section A.1. The supplement has also been updated with a notebook that replicates the results presented in that section - please refer to the README.md therein.

Please let us know if you have any further questions or comments!

---

### Meta-Review · Area_Chair_vBVM · 2023-12-11

**Metareview:**

This paper presents causal disentanglement with extreme values (CDEV), a novel method for identifying meaningful latent representations from unknown data, with applications to sperm whale communication.  The reviewers found the contribution to be novel and highly interesting. However, three of the four reviewers expressed significant concerns about the clarity, motivation, and about whether the experiments were sufficient to support the major application-specific claims about whale communication. One reviewer also expressed additional concerns about originality and the use of causal terminology. Unfortunately, the enthuasiasm of one champion reviewer was not sufficient to outweight the more moderate appraisal of the other three, and this paper cannot be accepted to ICLR in its current form.  The work nevertheless seems promising and I wish the authors the best of luck in revising it for publication elsewhere.

**Justification For Why Not Higher Score:**

The most thorough, cogent review came from the "3" reviewer, and it seems to me their concerns are substantial enough to warrant rejection for this one.  Unfortunately, the two more positive reviewers have low confidence.

**Justification For Why Not Lower Score:**

n/a

---

### Decision · Program_Chairs · 2024-01-16

Reject